# MicroRNA-33 inhibition ameliorates muscular dystrophy by enhancing skeletal muscle regeneration

Naoya Sowa[1,2], Takahiro Horie [ID][1✉], Yuya Ide[1], Osamu Baba[1], Kengo Kora[3], Takeshi Yoshida [ID][3], Yujiro Nakamura[4], Shigenobu Matsumura [ID][5], Kazuki Matsushita[1], Miyako Imanaka[1], Fuquan Zou[1], Eitaro Kume[1,3], Hidenori Kojima[1], Qiuxian Qian[1], Kayo Kimura[1,6], Ryotaro Otsuka[1,7], Noriko Hara[1], Tomohiro Yamasaki[1], Chiharu Otani[1], Yuta Tsujisaka[1], Tomohide Takaya [ID][8], Chika Nishimura[9], Dai Watanabe[9], Koji Hasegawa[2], Jun Kotera[10], Kozo Oka[10], Ryo Fujita[10], Akihiro Takemiya[10], Takashi Sasaki[10], Yuuya Kasahara [ID][11], Satoshi Obika[12], Takeshi Kimura[1] & Koh Ono [ID][1✉]

## Abstract

**Muscular dystrophy is a group of diseases characterized by progressive weakness and degeneration of skeletal muscles, for which there is currently no cure. Here, we show that microRNA (miR)-33a/b play a crucial role in muscle regeneration. miR-33a was upregulated during myoblast differentiation and in skeletal muscles of *mdx* mice, a genetic model of Duchenne muscular dystrophy (DMD). miR-33a deficiency enhanced muscle regeneration response to cardiotoxin injury and attenuated muscle degeneration and fibrosis in *mdx* mice. Conversely, a humanized mouse model expressing miR-33a and miR-33b showed exacerbated muscle degeneration and fibrosis. Mechanistically, miR-33a/b inhibited satellite cell proliferation, leading to reduced muscle regeneration and increased fibrosis by targeting *Cdk6*, *Fst*, and *Abca1*. Local and systemic administration of anti-miRNA oligonucleotides targeting miR-33a/b ameliorated the dystrophic phenotype in *mdx* mice. Furthermore, miR-33b inhibition upregulated these target genes in myotubes differentiated from human induced pluripotent stem cells derived from a patient with DMD. These findings indicate that miR-33a/b are involved in muscle regeneration and their inhibition may represent a potential therapeutic strategy for muscular dystrophy.**

**Keywords** Muscular Dystrophy; Regeneration; microRNA-33a; microRNA-33b; Antisense Oligonucleotide
**Subject Categories** Musculoskeletal System; RNA Biology

See also: MA Lopez and MS Alexander

## Introduction

Muscular dystrophy is a class of genetic diseases characterized by progressive skeletal muscle weakness and degeneration. Duchenne muscular dystrophy (DMD) is a severe and progressive form caused by the loss of function of the dystrophin gene (*DMD*) on the X chromosome (Hoffman et al, 1987). In contrast, patients with Becker muscular dystrophy (BMD) produce partially functional dystrophin and present a milder phenotype than DMD (Angelini et al, 1994). The absence of dystrophin leads to muscle fiber damage during contraction, resulting in accelerated inflammation and fibrosis that inhibits muscle regeneration (Duan et al, 2021). Although recent therapeutic advances, such as exon skipping (Echevarría et al, 2018; McDonald et al, 2021) or stop codon read-through (McDonald et al, 2017; Welch et al, 2007), have resulted in the production of partially functional dystrophin in specific subsets of DMD patients, no curative treatments for DMD are currently available in clinical practice (Duan et al, 2021; Markati et al, 2022). Furthermore, only a few types of supportive care, such as nutrition, rehabilitation, and respiratory support (including ventilation), slow disease progression and increase life expectancy, but their efficacy is

[1]Department of Cardiovascular Medicine, Graduate School of Medicine, Kyoto University, 54 Shogoin-kawahara-cho, Sakyo-ku, Kyoto 606-8507, Japan. [2]Division of Translational Research, National Hospital Organization, Kyoto Medical Center, 1-1 Fukakusa Mukaihata-cho, Fushimi-ku, Kyoto 612-8555, Japan. [3]Department of Pediatrics, Graduate School of Medicine, Kyoto University, 54 Shogoin-kawahara-cho, Sakyo-ku, Kyoto 606-8507, Japan. [4]Division of Food Science and Biotechnology, Graduate School of Agriculture, Kyoto University, Kitashirakawa Oiwake-cho, Sakyo-ku, Kyoto 606-8502, Japan. [5]Department of Nutrition, Osaka Metropolitan University, 3-7-30 Habikino, Osaka 600-8891, Japan. [6]Department of Anesthesia, Graduate School of Medicine, Kyoto University, 54 Shogoin-kawahara-cho, Sakyo-ku, Kyoto 606-8507, Japan. [7]Department of Neurosurgery, Graduate School of Medicine, Kyoto University, 54 Shogoin-kawahara-cho, Sakyo-ku, Kyoto 606-8507, Japan. [8]Department of Agriculture, Graduate School of Science and Technology, Shinshu University, 8304 Minami-minowa, Kami-ina, Nagano 399-4598, Japan. [9]Department of Biological Sciences, Graduate School of Medicine, Kyoto University, Yoshida, Sakyo-ku, Kyoto 606-8501, Japan. [10]Mitsubishi Tanabe Pharma Corporation, 1000 Kamoshida-cho, Aoba-ku, Yokohama 227-0033, Japan. [11]National Institutes of Biomedical Innovation, Health and Nutrition, 7-6-8, Saito-Asagi, Ibaraki City, Osaka 567-0085, Japan. [12]Graduate School of Pharmaceutical Sciences, Osaka University, 1-6 Yamadaoka, Suita, Osaka 565-0871, Japan. ✉E-mail: thorie@kuhp.kyoto-u.ac.jp; kohono@kuhp.kyoto-u.ac.jp

limited. Therefore, novel therapeutic approaches are crucial to treat patients with DMD.

MicroRNAs (miRNAs; miRs) are a class of short, noncoding RNAs that exert substantial influence on numerous biological pathways through posttranscriptional repression of their target genes. Several miRNAs are involved in regulating muscle differentiation, some of which modulate muscle regeneration. For instance, skeletal muscle-specific miR-1 and miR-206 regulate the proliferation and differentiation of satellite cells (SCs), which function as stem cells in skeletal muscle (Chen et al, 2010); miR-206-deficient mice show delayed regeneration and an exacerbated dystrophic phenotype (Liu et al, 2012); and the inhibition of dystrophin suppressors miR-146b, miR-374a, and miR-31 has a beneficial effect on muscular dystrophy (Fiorillo et al, 2015). Furthermore, miR-21 and miR-199a-5p suppression and miR-29 overexpression decelerate fibrosis, thereby promoting muscle regeneration (Wang et al, 2012; Zanotti et al, 2018; Zanotti et al, 2015).

We and other groups have shown that miR-33a, transcribed from the intron of sterol regulatory element-binding factor 2 gene (*SREBF2*), regulates lipid homeostasis in conjunction with its host genes by targeting ATP-binding cassette transporter A1 (*ABCA1*), which exports intracellular cholesterol to apolipoprotein A-I, leading to high-density lipoprotein cholesterol (HDL-C) formation (Horie et al, 2010a; Najafi-Shoushtari et al, 2010; Rayner et al, 2010). miR-33a-knockout (miR-33a-KO) mice or mice in which miR-33a is inhibited show reduced atherosclerosis with an improved serum cholesterol profile, enhanced reverse cholesterol transport, and ameliorated vascular inflammation (Horie et al, 2012; Rayner et al, 2011). Rodents have one type of miR-33 (miR-33a), transcribed from an intron of *Srebf2*. However, in humans, there is also miR-33 (miR-33b) transcribed from the intron of *SREBF1*, giving rise to two types of miR-33 (miR-33a and miR-33b) (Horie et al, 2014b). Humanized miR-33b knock-in (miR-33b-KI) mice, which have a miR-33b sequence in the same intron of *Srebf1* as humans, exhibit accelerated atherosclerosis in contrast to miR-33a-deficient mice (Nishino et al, 2018). Therefore, miR-33a/b are potential therapeutic targets for dyslipidemia and atherosclerosis (Horie et al, 2014a).

miR-33a/b are relatively abundant in human skeletal muscles. We observed miR-33a upregulation during myoblast differentiation and in *mdx* mice (a genetic mouse model of DMD). Thus, we hypothesized that miR-33 influenced the regeneration of damaged muscles, and analyzed multiple genetically modified mouse models. miR-33a-KO mice showed enhanced muscle regeneration and reduced fibrosis in both cardiotoxin (CTX)-induced muscle injury and *mdx* mouse models. In contrast, miR-33b-KI *mdx* mice showed reduced muscle regeneration and exacerbated fibrosis. Several miR-33 target genes such as *Cdk6*, *Fst* and *Abca1* were identified as responsible for these phenotypes. We also developed anti-microRNA oligonucleotides (AMOs) that effectively inhibited miR-33a/b, as evidenced by the successful amelioration of the dystrophic phenotype and exercise capacity in *mdx* mice following local and systemic AMO administration. Moreover, miR-33b inhibition upregulated these miR-33 target genes in myotubes differentiated from human induced pluripotent stem (iPS) cells derived from a patient with DMD. Our findings indicate the involvement of miR-33a/b in muscle regeneration and suggest its potential as a therapeutic target for muscular dystrophy, including DMD.

# Results

## miR-33a is abundantly expressed in skeletal muscle

We used a commercially available RNA panel to quantify miR-33a expression in various human organs. miR-33a was abundant in skeletal muscle and expressed at equivalent levels in the liver, which corresponded with the levels of its host gene, *SREBF2* (Appendix Fig. S1A) (Koyama et al, 2019). Then, we evaluated miR-33a and *Srebf2* levels in several organs of wild-type (WT) mice. The levels in the gastrocnemius (GAS) muscle were comparable to those in the liver (Appendix Fig. S1B), and the levels in the skeletal muscle were similar in fast-twitch muscles, such as the GAS and tibialis anterior (TA) muscle, and in slow-twitch muscles, such as the soleus muscle (SOL) (Appendix Fig. S1C). Moreover, miR-33a and *Srebf2* expression were upregulated during the myogenic differentiation of mouse myoblast C2C12 cells (Appendix Fig. S1D).

## miR-33a deficiency accelerates skeletal muscle regeneration after CTX injection

We previously generated miR-33a-KO mice (Horie et al, 2010a). These mice exhibited no apparent locomotor abnormalities or histological differences in the TA muscle by hematoxylin and eosin (HE) staining and lectin staining under steady-state conditions (Appendix Fig. S2A,B). Because miR-33a expression was upregulated during C2C12 differentiation and it has been reported that miR-33a regulates cell proliferation and cell cycle progression in hepatocytes (Cirera-Salinas et al, 2012), we hypothesized that miR-33a influenced skeletal muscle regeneration. Accordingly, we evaluated the regenerative response of the TA muscle in WT and miR-33a-KO mice following CTX injection, which induces myolysis and triggers muscle regeneration (Guardiola et al, 2017). Seven days after CTX injection, we observed degenerated muscle fibers in both WT and miR-33a-KO mice. However, the extent of degeneration was less severe in miR-33a-KO mice compared with WT mice (Appendix Fig. S2C). We also stained TA muscle sections with an antibody against Pax7, a marker of muscle SCs, revealing that the number of Pax7[+] cells was higher in miR-33a-KO mice than WT mice (Appendix Fig. S2B,D). Fourteen days after CTX injection, most of the damaged myofibers had been replaced with regenerated myofibers in both groups. Pax7[+] cells had disappeared in miR-33a-KO mice but were still present in WT mice (Appendix Fig. S2C,D). These findings indicated that miR-33a-KO mice showed accelerated muscle regeneration (completed in 14 days) after CTX injection. There was no difference in muscle weight between WT and miR-33a-KO mice after phosphate-buffered saline (PBS) injection. Muscle weight tended to increase after CTX injection, showing a significant difference in miR-33a-KO mice at 21 days after CTX injection (Appendix Fig. S2E). miR-33a expression was upregulated during the recovery period following the CTX injection (Appendix Fig. S2F). miR-33 has been reported to regulate hepatocyte proliferation by targeting cyclin-dependent kinase 6 (*CDK6*) and cyclin D1 (*CCND1*), which enhance cell proliferation and cell cycle progression (Cirera-Salinas et al, 2012). Therefore, we measured the expression of these genes in the TA muscle. The protein levels of ABCA1, a well-established target gene of miR-33, and CDK6 were upregulated, whereas the level of CCND1 remained unchanged in the TA muscle of miR-33a-KO

mice compared with WT mice (Appendix Fig. S3A). The baseline levels of other myogenic genes did not differ significantly between the WT and miR-33a-KO mice (Appendix Fig. S3B). Therefore, miR-33 appears to regulate CDK6 at the translational level rather than affecting mRNA stability.

## miR-33a deficiency ameliorates dystrophic phenotypes in *mdx* mice

Next, we investigated the function of miR-33a in *mdx* mice, a mouse model of DMD. These mice harbor a nonsense point mutation in exon 23 of *Dmd*, resulting in a lack of full-length dystrophin expression (Cox et al, 1993). miR-33a expression was elevated in the TA muscle of *mdx* mice compared with that of WT mice (Appendix Fig. S4A). We crossed miR-33a-KO mice with *mdx* mice and compared the muscle phenotypes between 8-week-old WT/*mdx* and miR-33a-KO/*mdx* (KO/*mdx*) mice. Because the miR-33a-KO mice had a C57/BL6 background and *mdx* mice had a C57/BL10 background, littermate WT/*mdx* mice were used as controls. The KO/*mdx* mice tended to have heavier body weight than the WT/*mdx* mice, although their muscle weights were comparable (Appendix Fig. S4B,C). However, the size of myofibers with centralized nuclei, which indicates regenerating myofibers, was significantly larger in the TA muscle of KO/*mdx* mice than that of WT/*mdx* mice, as observed by HE staining (Fig. 1A,B). The fibrotic area, measured by Masson trichrome staining, was significantly smaller in the TA muscle of KO/*mdx* mice compared with WT/*mdx* mice (Fig. 1C,D). The level of serum creatinine kinase (CK), a marker of muscle damage, was significantly lower in 4-week-old KO/*mdx* mice compared with WT/*mdx* mice (Fig. 1E). We performed a wire-hanging test and treadmill test to assess exercise endurance. KO/*mdx* mice could show higher body mass (g) × hanging time (s) (holding impulse) (Fig. 1F) and run for a longer time and greater distance than WT/*mdx* mice (Fig. 1G–I). These findings indicated that miR-33a deficiency ameliorated the dystrophic phenotype in *mdx* mice.

## miR-33a regulates SC proliferation

We analyzed and characterized the TA muscle of KO/*mdx* mice. Myogenic differentiation markers, such as *Myog* and *Myod1*, were elevated in the TA muscle of KO/*mdx* mice (Fig. 1J), *Pax7* was upregulated in KO/*mdx* mice at both mRNA and protein levels (Fig. 1J; Appendix Fig. S4D), and the mRNA and protein levels of CDK6 were significantly upregulated in KO/*mdx* mice (Fig. 1J–L) compared with WT/*mdx* mice. The protein expression of downstream target genes of CDK6, including phosphorylated retinoblastoma protein (Rb) and proliferating cell nuclear antigen (PCNA), were increased in the muscles of KO/*mdx* mice compared with WT/*mdx* mice (Appendix Fig. S4E). In line with these findings, CDK6 expression in the extensor digitorum longus (EDL) muscle was upregulated in WT/*mdx* mice and further increased in KO/*mdx* mice (Appendix Fig. S4F). Double immunostaining revealed a significant increase in the number of Pax7+/MyoD− and Pax7+/MyoD+ cells (indicating quiescent and activated SCs, respectively) (Tedesco et al, 2010) in the TA muscle of KO/*mdx* mice compared with WT/*mdx* mice (Fig. 2A,B). Furthermore, the number of Pax7+/Ki67+ cells increased in the muscles of KO/*mdx* mice compared with WT/*mdx* mice (Fig. 2C,D). These data

suggested that SCs were more abundant and proliferative in KO/*mdx* mice than in WT/*mdx* mice.

Utrophin, a structural and functional paralog of dystrophin (80% homology) is functionally redundant (Guiraud and Davies, 2017). Utrophin levels are known to be increased in *mdx* mice and DMD patients to compensate for the lack of dystrophin (Anthony et al, 2014; Tinsley et al, 1996). Our western blotting and immunohistochemistry results revealed that utrophin expression was increased in the muscle of KO/*mdx* mice compared with WT/*mdx* mice (Fig. 2E–G).

We isolated primary myoblasts from the EDL and TA muscles (fast-twitch muscles) of WT/*mdx* and KO/*mdx* mice. The proliferation rates were higher in KO/*mdx* mice compared with WT/*mdx* mice (Appendix Fig. S5A,B), and this difference disappeared following the introduction of shRNA against *Cdk6* using a lentivirus vector (Appendix Fig. S5C,D). Upon differentiation into myotubes, the number of mature myotubes was higher in the primary myoblasts from KO/*mdx* mice than in those of WT/*mdx* mice (Appendix Fig. S5E), as confirmed by the fusion index (ratio of nuclei in myotubes to total nuclei) (Appendix Fig. S5F,G). During myotube differentiation, miR-33a expression increased in a manner similar to that observed in C2C12 cells (Appendix Figs. S2F and S5H). Furthermore, compared with WT/*mdx* mice, utrophin levels were elevated in the differentiated myotubes from KO/*mdx* mice (Appendix Fig. S5I). Thus, SCs and myoblasts in KO/*mdx* mice exhibited higher proliferation rates with increased *Cdk6* expression, contributing to myotube maturation and potentially leading to the amelioration of exercise endurance and fibrosis in KO/*mdx* mice.

## miR-33b-KI shows the opposite phenotype to miR-33a deficiency in *mdx* mice

As a gain-of-function model, we analyzed humanized miR-33b-KI mice, in which the human miR-33b sequence was inserted within the same intron of *Srebf1* as in humans (Horie et al, 2014b). Although rodents have only one miR (miR-33a), miR-33b-KI mice express two miRs (miR-33a and miR-33b), like humans. In these mice, miR-33b is physiologically coexpressed with its host gene, *Srebf1*, and is abundantly expressed in muscle tissue (Koyama et al, 2019). We found that miR-33b-KI mice did not show any apparent changes in the TA muscle histology (Appendix Fig. S6A,B). Compared with WT mice, the expression levels of *Myog*, *Myod1*, and *Pax7* were comparable, whereas *Cdk6* expression was significantly reduced in the muscles of miR-33b-KI mice (Appendix Fig. S6C). We generated miR-33b-KI/*mdx* mice by crossing miR-33b-KI mice with *mdx* mice and used littermates as controls (WT/*mdx*). Although the body weight remained unchanged (Appendix Fig. S6D), the weight of the TA muscle was significantly lower in miR-33b-KI/*mdx* mice than in WT/*mdx* mice (Appendix Fig. S6E,F). Histological analysis revealed that the regenerating muscle fibers in miR-33b-KI/*mdx* mice were smaller (Fig. 3A,B) and the fibrotic area was significantly larger in miR-33b-KI/*mdx* mice than in WT/*mdx* mice (Fig. 3C,D). Furthermore, serum CK levels were significantly elevated in miR-33b-KI/*mdx* mice compared with WT/*mdx* mice (Fig. 3E). The wire-hanging and treadmill endurance tests revealed reduced exercise capacity in miR-33b-KI/*mdx* mice compared with WT/*mdx* mice (Fig. 3F–H). Compared with the WT/*mdx* mice, miR-33b-KI/*mdx* mice showed reduced expression levels of *Myog*, *Myod1*, *Pax7*, and *Cdk6* (Fig. 3I)

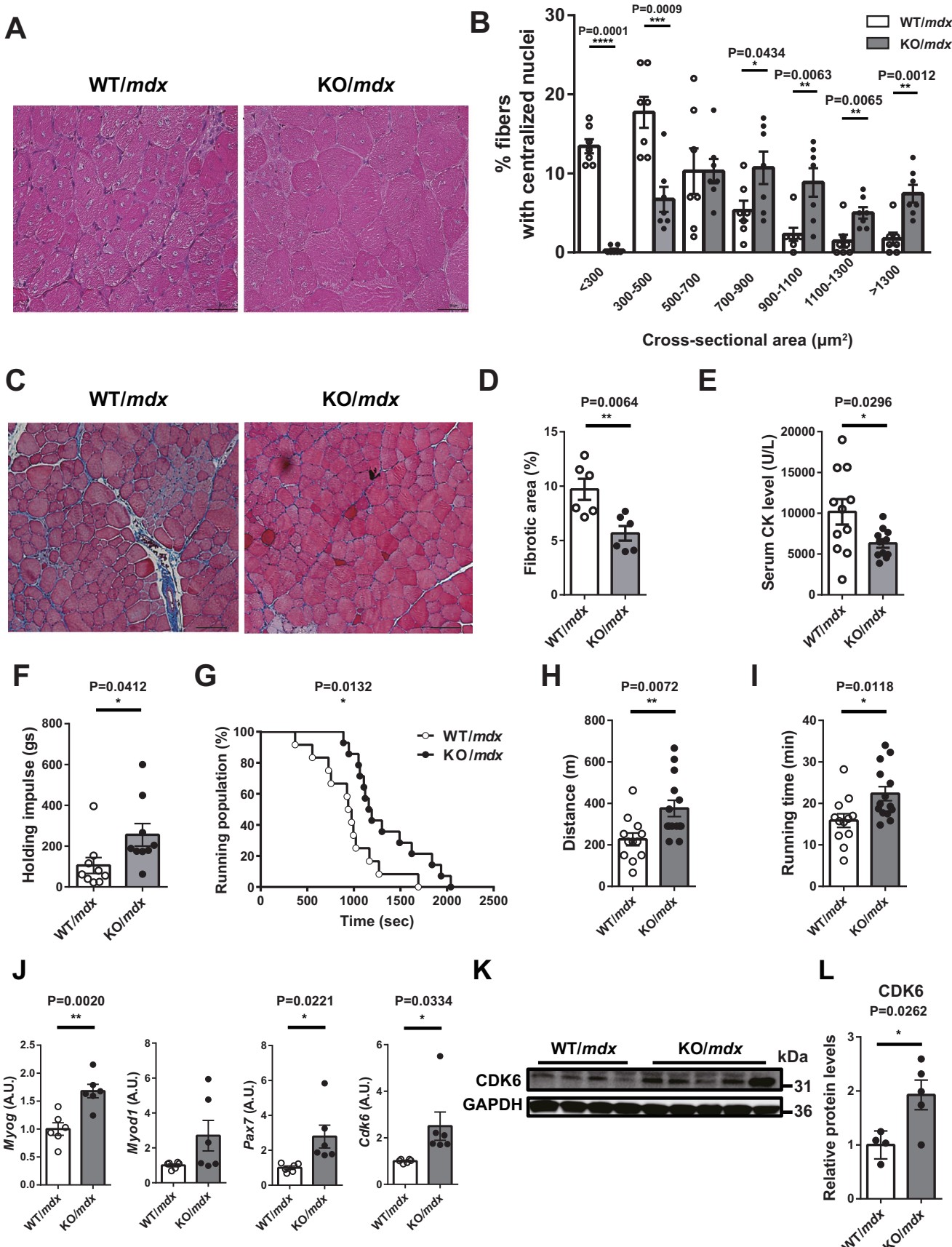

**Figure 1.   miR-33a deficiency ameliorates the dystrophic phenotype in *mdx* mice.**

(A) Representative images of HE staining of the TA muscle of WT/*mdx* and KO/*mdx* mice. Scale bar: 50 μm. (B) Size distribution of muscle fibers in the TA muscle of WT/*mdx* and KO/*mdx* mice (*n* = 7/group). Unpaired *t* test. (C) Representative images of Masson trichrome staining of TA muscle of WT/*mdx* and KO/*mdx* mice. Scale bar: 100 μm. (D) Percentage of fibrotic area in TA muscle of WT/*mdx* and KO/*mdx* mice (*n* = 6/group). Unpaired *t* test. (E) Serum CK levels in WT/*mdx* and KO/*mdx* mice (*n* = 11/group). Unpaired *t* test. (F) Wire-hanging test in WT/*mdx* and KO/*mdx* mice (*n* = 9/group). Unpaired *t* test. (G) Running population plotted against time during the treadmill endurance test in WT/*mdx* (*n* = 12) and KO/*mdx* mice (*n* = 14). Log-rank test. (H) Running distance in the treadmill endurance test for WT/*mdx* (*n* = 12) and KO/*mdx* (*n* = 14) mice. Unpaired *t* test. (I) Running time in the treadmill endurance test for WT/*mdx* mice (*n* = 12) and KO/*mdx* (*n* = 14) mice. Unpaired *t* test. (J) *Myog*, *Myod1*, *Pax7*, and *Cdk6* expression in the TA muscle of WT/*mdx* and KO/*mdx* mice (*n* = 6/group). Unpaired *t* test. (K) Western blotting for CDK6 and GAPDH proteins in TA muscle of WT/*mdx* and KO/*mdx* mice. (L) Densitometric analysis of CDK6 in the TA muscle of WT/*mdx* (*n* = 4) and KO/*mdx* (*n* = 5) mice. Unpaired *t* test. Data are presented as the mean ± SEM. *$P < 0.05$, **$P < 0.01$, ***$P < 0.001$, ****$P < 0.0001$. Source data are available online for this figure.

and suppressed protein levels of ABCA1, PAX7, and CDK6 (Fig. 4A,B). The number of Pax7$^+$ cells, particularly Pax7$^+$/MyoD$^-$ cells, was decreased in the TA muscle of miR-33b-KI/*mdx* mice compared with WT/*mdx* mice (Fig. 4C,D). Western blotting and immunohistochemistry results showed that utrophin expression was reduced in miR-33b-KI/*mdx* mice compared with WT/*mdx* mice (Fig. 4E–G). Furthermore, primary myoblast proliferation was suppressed and myotube maturation was impaired in miR-33b-KI/*mdx* mice compared with WT/*mdx* mice (Appendix Fig. S6G–J). These findings indicated that SCs and myoblasts in miR-33b-KI/*mdx* mice showed lower proliferation rates with reduced *Cdk6* expression compared with miR-33a-KO/*mdx* mice, resulting in an exacerbated dystrophic phenotype.

### *Cdk6* knockdown reverses the beneficial phenotype of miR-33a deficiency *in mdx* mice

The above-mentioned findings suggested that *Cdk6* was responsible for the muscle phenotype observed in miR-33a-KO/*mdx* and miR-33b-KI/*mdx* mice. To confirm the molecular link between miR-33a/b and the muscle phenotype in vivo, we conducted a rescue experiment using AAV9. We introduced shRNA against *Cdk6* (sh*Cdk6*) into the TA muscle of 8-week-old miR-33a-KO/*mdx* mice by direct injection of AAV9 and sacrificed them for analysis after 28 days (Fig. 5A). We confirmed the transduction efficacy of AAV9 into the TA muscle by injecting an AAV9 vector containing a CMV-driven green fluorescent protein (GFP) cassette (Appendix Fig. S7A). Because the shRNA AAV9 vector harbors a CMV-driven dsRed expression cassette along with a U6-driven shRNA expression cassette, the infected muscle tissue had a reddish color (Fig. 5B). CDK6 expression was effectively suppressed by injection of the sh*Cdk6* AAV9 vector (Fig. 5C,D). Compared with the shRNA control AAV9 vector, injection of the sh*Cdk6* AAV9 vector significantly downregulated *Myod1*, *Myog*, and *Pax7* expression (Fig. 5C), and increased the fibrotic area in TA muscle of miR-33a-KO/*mdx* mice (Fig. 5E,F). SCs, particularly Pax7$^+$/MyoD$^-$ cells, were decreased in the sh*Cdk6* AAV9-infected TA muscle of miR-33a-KO/*mdx* mice (Fig. 5G,H). Immunohistochemistry and western blotting revealed attenuated utrophin expression in the sh*Cdk6* AAV9-infected TA muscle of miR-33a-KO/*mdx* mice (Appendix Fig. S7B,C), and PAX7 expression was also suppressed in this group (Appendix Fig. S7C).

Because *Abca1* expression was also altered in miR-33a-KO/*mdx* and miR-33b-KI/*mdx* mice, we investigated the effect of *Abca1* knockdown on the TA muscle of miR-33a-KO/*mdx* mice (Appendix Fig. S7D,E). *Myod1*, *Myog*, and *Pax7* expression remained unchanged, but the fibrotic area increased in sh*Abca1*

AAV9-infected TA muscle (Appendix Fig. S7E–G). These findings indicated that *Cdk6* upregulation and partially upregulation of *Abca1* were responsible for the favorable muscle phenotype in miR-33a-KO/*mdx* mice.

### *FST* is another target gene of miR-33 and is responsible for the miR-33a deficiency phenotype

We further investigated additional target genes of miR-33 involved in the muscle phenotype of miR-33a-KO/*mdx* and miR-33b-KI/*mdx* mice. Follistatin (FST) is a member of the transforming growth factor-β (TGF-β) superfamily that inhibits the function of myostatin, a negative regulator of muscle development and growth (Chen and Lee, 2016). FST promotes muscle growth through SC proliferation (Gilson et al, 2009; Yaden et al, 2014) or muscle healing and regeneration (Zhu et al, 2011). We identified a potential miR-33-binding site in the 3′-UTR of *FST*, which was highly conserved among species (Appendix Fig. S8A,B). We confirmed that miR-33 targeted *FST* using an *FST* 3′-UTR-luciferase assay. miR-33a significantly suppressed luciferase activity in the WT *FST* 3′-UTR, and this suppression was reversed by introducing mutations into the miR-33-binding site (Appendix Fig. S8C). FST expression was increased in the muscles of miR-33a-KO mice (Appendix Fig. S8D,E) and miR-33a-KO/*mdx* mice (Appendix Fig. S8F–H) but was decreased in miR-33b-KI/*mdx* mice at both the mRNA and protein levels (Appendix Fig. S8I–K). We performed in vivo rescue experiments for miR-33a-KO/*mdx* mice using AAV9-mediated shRNA against *Fst* (Fig. 6A), as in the case of *Cdk6* and *Abca1*. The sh*Fst* AAV9-infected TA muscle of miR-33a-KO/*mdx* mice showed reduced expression of FST and myogenic genes (Fig. 6B,C), an increased area of fibrosis (Fig. 6D,E), and decreased Pax7$^+$/MyoD$^-$ cells (Fig. 6F,G). Furthermore, utrophin expression was diminished (Appendix Fig. S8L), and PAX7 and CDK6 expression were suppressed (Appendix Fig. S8M). These findings indicated that *Fst* was another miR-33-target gene whose elevation contributed to the beneficial phenotype in miR-33a-KO/*mdx* mice.

### miR-33 inhibition ameliorates the dystrophic phenotype in *mdx* mice

To translate our findings into therapeutic applications, we developed AMOs using amido-bridged nucleic acid (AmNA)-based artificial nucleotides, which effectively suppress miR-33a and miR-33b function (Miyagawa et al, 2023; Yamasaki et al, 2022). We directly injected control anti-miRNA oligonucleotides (control AmNA) or anti-miRNA oligonucleotides against miR-33a (anti-

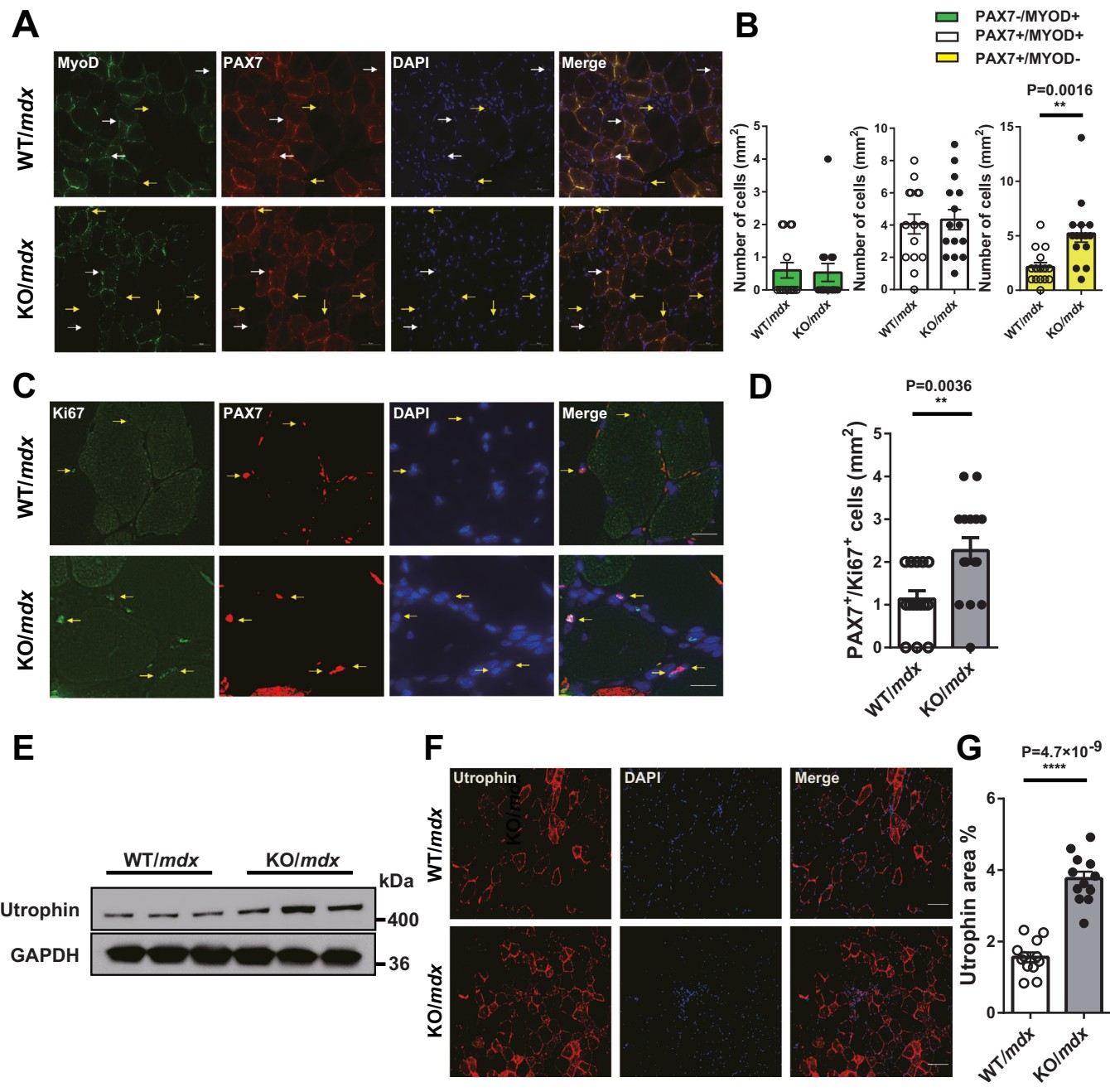

**Figure 2.  miR-33a regulates SC proliferation.**

(**A**) Representative fluorescent images of TA muscle from WT/*mdx* and KO/*mdx* mice stained with MyoD, Pax7, and DAPI. White and yellow arrows indicate Pax7+/MyoD+, and Pax7+/MyoD− cells, respectively. Scale bar: 50 µm. (**B**) Number of Pax7−/MyoD+, Pax7+/MyoD+, and Pax7+/MyoD− cells in the TA muscle of WT/*mdx* and KO/*mdx* mice ($n = 5$/group). Unpaired *t* test. (**C**) Representative fluorescent images of the TA muscle from WT/*mdx* and KO/*mdx* mice stained with Ki67, Pax7, and DAPI. Arrows indicate Pax7+/Ki67+ cells. Scale bar: 20 µm. (**D**) Number of Pax7+/Ki67+ cells in the TA muscle of WT/*mdx* and KO/*mdx* mice. Three fields of view/mouse ($n = 5$/group). Unpaired *t* test. (**E**) Western blotting for utrophin and GAPDH in TA muscle of WT/*mdx* and KO/*mdx* mice. (**F**) Representative fluorescent images of TA muscle from WT/*mdx* and KO/*mdx* mice stained with utrophin and DAPI. Scale bar: 50 µm. (**G**) Quantification of the utrophin-positive area in the TA muscle from WT/*mdx* and KO/*mdx* mice ($n = 12$/group). Unpaired *t* test. Data are presented as the mean ± SEM. **$P < 0.01$, ***$P < 0.001$, ****$P < 0.0001$. Source data are available online for this figure.

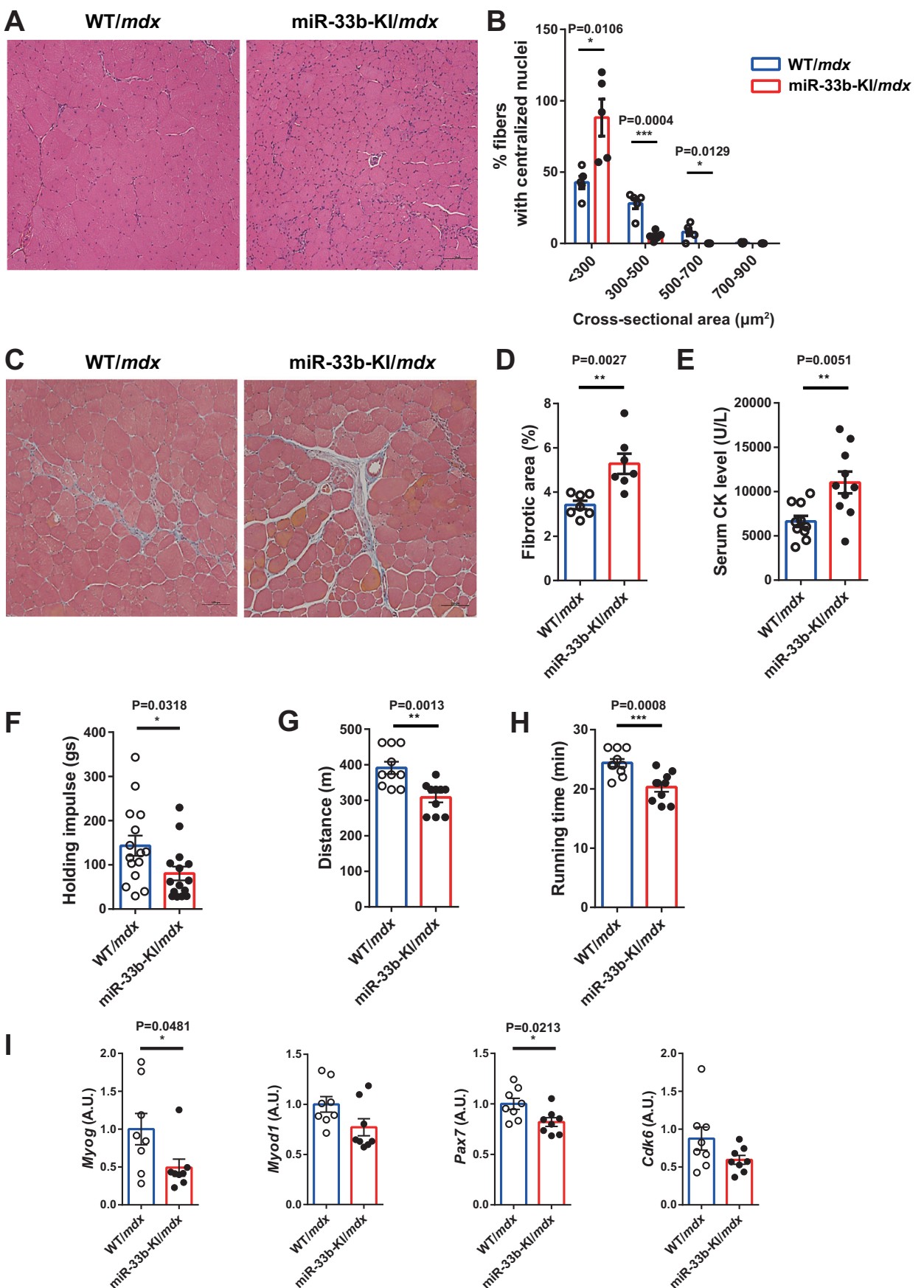

**Figure 3.  miR-33b-KI shows a phenotype opposite to that of miR-33a-KO in *mdx* mice.**

(A) Representative images of HE staining of the TA muscle of WT/*mdx* and miR-33b-KI/*mdx* mice. Scale bar: 100 μm. (B) Size distribution of TA muscle fibers in WT/*mdx* and miR-33b-KI/*mdx* mice. Three fields of view/mouse ($n = 5$/group). Unpaired $t$ test. (C) Representative images of Masson trichrome staining of the TA muscle of WT/*mdx* and miR-33b-KI/*mdx* mice. Scale bar: 100 μm. (D) Percentage of fibrotic area in TA muscle of WT/*mdx* and miR-33b-KI/*mdx* mice ($n = 7$/group). Unpaired $t$ test. (E) Serum CK levels in WT/*mdx* and miR-33b-KI/*mdx* mice ($n = 10$/group). Unpaired $t$ test. (F) Wire-hanging test in WT/*mdx* and miR-33b-KI/*mdx* mice ($n = 15$/group). Unpaired $t$ test. (G) Running distance in treadmill endurance test for WT/*mdx* and miR-33b-KI/*mdx* mice ($n = 10$/group). Unpaired $t$ test. (H) Running time in treadmill endurance test for WT/*mdx* and miR-33b-KI/*mdx* mice ($n = 10$/group). Unpaired $t$ test. (I) *Myog*, *Myod1*, *Pax7*, and *Cdk6* expression in the TA muscle of WT/*mdx* and miR-33b-KI/*mdx* mice ($n = 8$/group). Unpaired $t$ test. Data are presented as the mean ± SEM. *$P < 0.05$, **$P < 0.01$, ***$P < 0.001$. Source data are available online for this figure.

miR-33a) or miR-33b (anti-miR-33b) into the TA muscle of miR-33b-KI/*mdx* mice (50 μg/muscle) and analyzed the effects 3 days later. miR-33a and miR-33b expression were significantly suppressed by anti-miR-33a and anti-miR-33b injection, respectively, demonstrating the effective suppression of miR-33a and miR-33b in the skeletal muscle (Fig. 7A). Consequently, anti-miR-33a/b, particularly anti-miR-33b, effectively upregulated miR-33 target genes, such as *Abca1*, *Cdk6*, and *Fst*, along with myogenic genes, such as *Pax7*, *Myod1*, and *Utrn* (Fig. 7A). We then injected anti-miR-33b or control AmNA into the TA muscle of miR-33b-KI/*mdx* mice once a week for 4 weeks and analyzed its therapeutic effects. The size of the regenerating muscle fibers was larger (Fig. 7B,C) and the fibrotic area was significantly reduced (Fig. 7D,E) in the muscle injected with anti-miR-33b compared with the muscle injected with control AmNA. Immunostaining revealed that utrophin expression was elevated in the muscle injected with anti-miR-33b (Fig. 7F). The number of Pax7+ cells, particularly Pax7+/MyoD− cells, was increased in the TA muscle injected with anti-miR-33b (Fig. 7G,H). Additionally, anti-miR-33a treatment in *mdx* mice, which only have miR-33a, upregulated miR-33-target genes and myogenic genes at both the mRNA and protein levels by effectively inhibiting miR-33a (Appendix Fig. S9A,B). The size of the regenerating muscle fibers was larger, the fibrotic area was reduced, and utrophin levels were elevated in the anti-miR-33a group (Appendix Fig. S9C–G).

Next, we examined the efficacy of systemic AMO administration in addition to that of local AMO administration. Since the therapeutic effect of local administration of anti-miR-33b was greater than that of anti-miR-33a in miR-33b-KI/*mdx* mice, control AmNA or anti-miR-33b was injected subcutaneously into miR-33b-KI/*mdx* mice (20 mg/kg body weight [bw]) once a week for 4 weeks. The body weight was comparable between the control AmNA and anti-miR-33b groups (Appendix Fig. S10A), and miR-33b expression in muscle was effectively suppressed using this protocol (Appendix Fig. S10B). Systemic anti-miR-33b treatment upregulated miR-33 target genes, such as ABCA1, CDK6, and FST, and myogenic genes, such as PAX7 and utrophin (Fig. 8A; Appendix Fig. S10C). Furthermore, anti-miR-33b treatment reduced the serum levels of muscle-derived enzymes, such as CK, aspartate aminotransferase, and lactate dehydrogenase, and increased HDL-C levels (Table 1). The treadmill endurance test results showed that anti-miR-33b treatment ameliorated the decline in exercise capacity that was markedly observed in the control AmNA group (Fig. 8B). The size of the regenerating muscle fibers was larger (Fig. 8C; Appendix Fig. S10D), the fibrotic area in the muscle was reduced (Fig. 8D,E), the number of Pax7+ cells was increased (Fig. 8F; Appendix Fig. S10E), and utrophin levels were increased in anti-miR-33b-treated miR-33b-KI/*mdx* mice (Fig. 8G).

Similar trends were observed at a lower dose of 10 mg/kg bw (Appendix Fig. S11A–F and Appendix Table S1). We performed RNA sequence analysis to clarify global changes in gene expression in muscles systemically treated with control AmNA or anti-miR-33b (20 mg/kg bw). Hierarchical clustering heatmap of differentially expressed genes (DEGs) and the function of DEGs are shown in Fig. 9A,B. Pathway analysis revealed significant alterations in several pathways, including cell cycle (Fig. 9C). Among the upregulated genes following anti-miR-33b treatment, 37 genes were identified as miR-33 target genes by TargetScan database, in silico miRNA target prediction software (https://www.targetscan.org/vert_80/) (Fig. 9D; Appendix Table S2). Representative DEGs associated with miR-33 targets, cell cycle regulation, SC activation and proliferation, muscle regeneration, and muscle maturation are shown in Fig. 9E.

Finally, we evaluated the relevance of these findings in human diseases. A human iPS cell line derived from a patient with DMD (CiRA00111; deletion of exon 44, dystrophin gene) (Li et al, 2015) was differentiated into myotube using a Tet-inducible MyoD expression system (Appendix Fig. S12A) (Uchimura et al, 2017; Yoshida et al, 2017). These cells were treated with control AmNA or anti-miR-33b at a dose of 100 nM. Efficient miR-33b suppression was observed following anti-miR-33b treatment (Appendix Fig. S12B). Anti-miR-33b treatment upregulated miR-33 target genes including CDK6, FST, and ABCA1, as well as utrophin in myotubes differentiated from DMD patient-derived iPS cells, consistent with the observation in mice (Appendix Fig. S12B; Fig. 9F,G). These findings suggested that miR-33 inhibition ameliorated the dystrophic phenotype by increasing several miR-33 target genes associated with cell cycle and muscle regeneration and might be a novel therapeutic approach for muscular dystrophy.

## Discussion

In the current study, we used various miR-33 genetically modified mice to elucidate the substantial role that miR-33 plays in skeletal muscle regeneration. miR-33a-KO mice showed accelerated muscle regeneration in response to CTX injury and ameliorated the dystrophic phenotype in *mdx* mice. In contrast, as a gain-of-function, miR-33b-KI mice exacerbated the dystrophic phenotype in *mdx* mice. Pax7+ cells were enriched in the muscle of miR-33a-KO/*mdx* mice but diminished in the muscle of miR-33b-KI/*mdx* mice. Mechanistically, miR-33 regulates the cell cycle of SCs and fibrosis by targeting the 3′-UTR of *Cdk6*, *Fst*, and partially, *Abca1*. We observed that AAV9-mediated shRNA reversed the amelioration of the dystrophic phenotype in the muscles of miR-33a-KO/*mdx* mice by reducing *Cdk6*, *Fst*, and *Abca1* expression. We

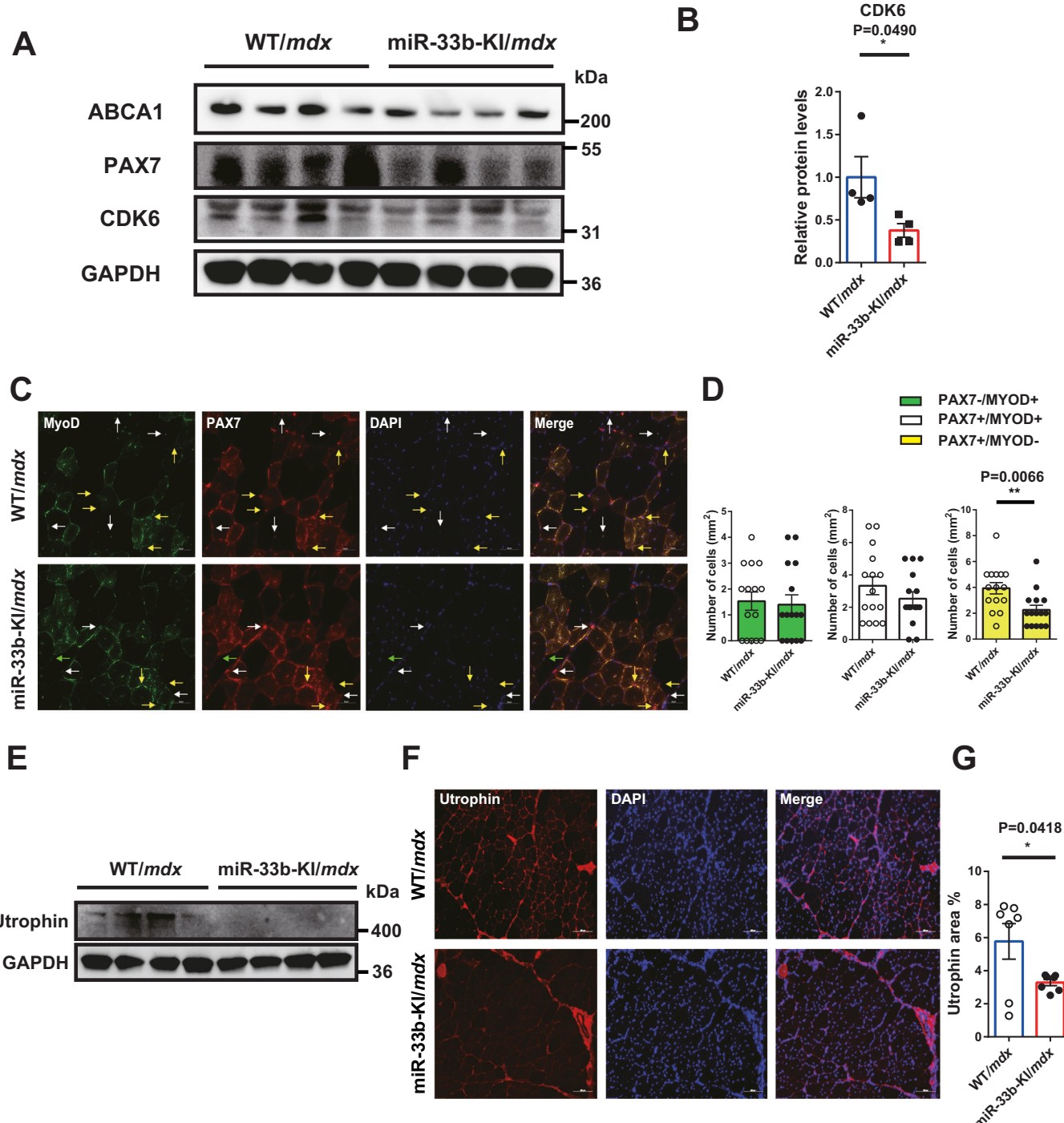

**Figure 4. miR-33b-KI reduces the SC proliferation.**

(A) Western blotting for ABCA1, Pax7, CDK6, and GAPDH in the TA muscle of WT/*mdx* and miR-33b-KI/*mdx* mice. (B) Densitometric analysis of CDK6 levels in the TA muscle of WT/*mdx* and miR-33b-KI/*mdx* mice (n = 4/group). Unpaired *t* test. (C) Representative fluorescent images of TA muscle from WT/*mdx* and miR-33b-KI/*mdx* mice stained with MyoD, Pax7, and DAPI. Green, white and yellow arrows indicate Pax7−/MyoD+, Pax7+/MyoD+, and Pax7+/MyoD− cells, respectively. Scale bar: 50 μm. (D) Number of Pax7−/MyoD+, Pax7+/MyoD+, and Pax7+/MyoD− cells in the TA muscle of WT/*mdx* and miR-33b-KI/*mdx* mice. Three fields of view/mouse (n = 5/group). Unpaired *t* test. (E) Western blotting for utrophin and GAPDH in the TA muscle of WT/*mdx* and miR-33b-KI/*mdx* mice. (F) Representative fluorescent images of TA muscle from WT/*mdx* and miR-33b-KI/*mdx* mice stained with utrophin and DAPI. Scale bar: 50 μm. (G) Quantification of the utrophin-positive area in TA muscle from WT/*mdx* and miR-33b-KI/*mdx* mice (n = 7/group). Unpaired *t* test. Data are presented as the mean ± SEM. *P < 0.05, **P < 0.01. Source data are available online for this figure.

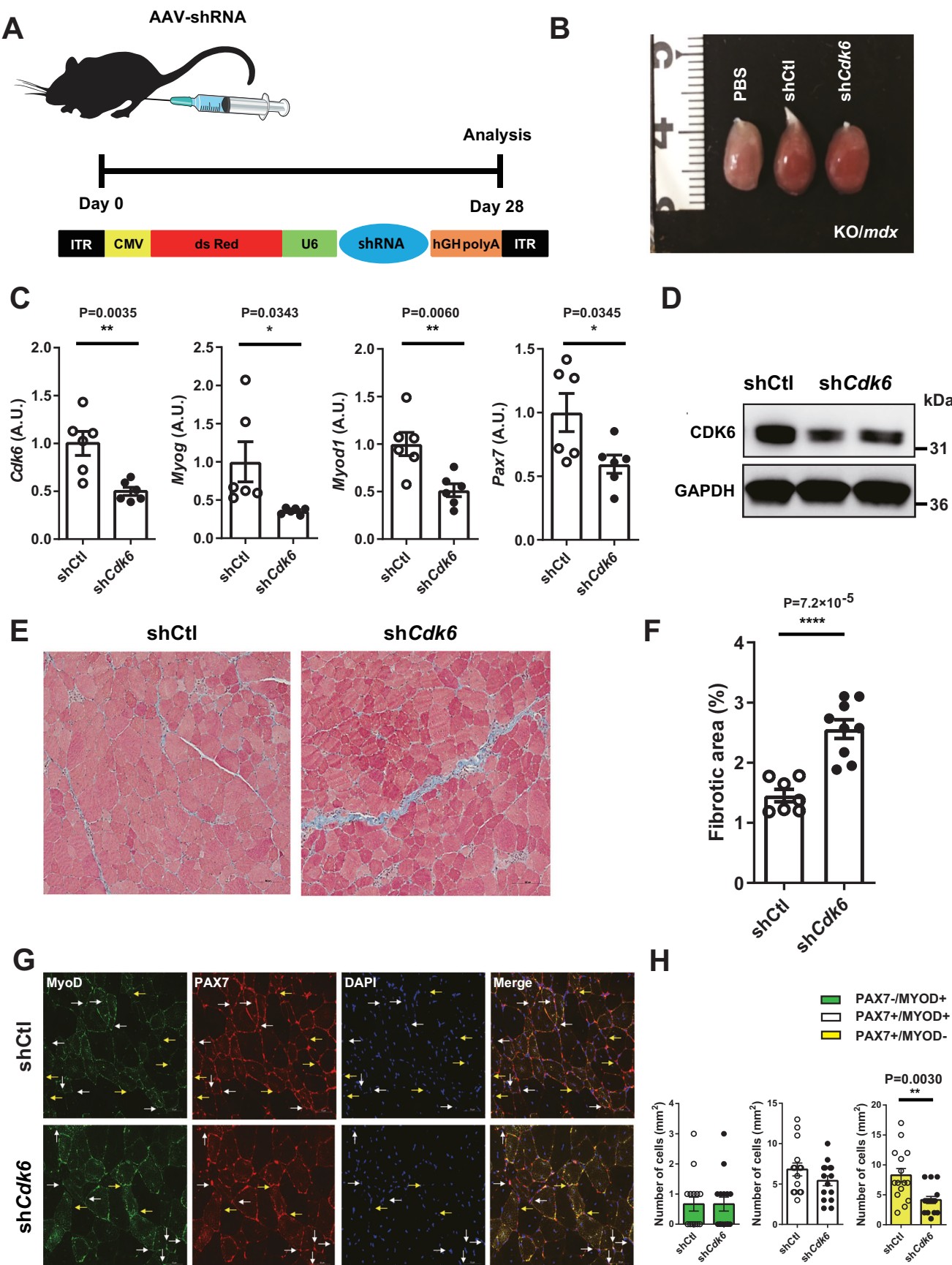

**Figure 5. *Cdk6* knockdown reverses the beneficial phenotype of miR-33a-KO/*mdx* mice.**

(A) Scheme of AAV9-mediated rescue experiment with shRNA AAV9 vector. (B) Representative images of TA muscle from KO/*mdx* mice injected with PBS, AAV9 shRNA control (shCtl), or AAV9 shRNA against *Cdk6* (sh*Cdk6*). (C) Expression of *Cdk6*, *Myog*, *Myod1*, and *Pax7* in the TA muscle of KO/*mdx* mice injected with AAV9 shCtl or sh*Cdk6* (n = 6/group). Unpaired *t* test. (D) Western blotting for CDK6 and GAPDH in the TA muscle of KO/*mdx* mice injected with AAV9 shCtl or sh*Cdk6*. (E) Representative images of Masson trichrome staining of TA muscle of KO/*mdx* mice injected with AAV9 shCtl or sh*Cdk6*. Scale bar: 100 μm. (F) Percentage of fibrotic area in the TA muscle of KO/*mdx* mice injected with AAV9 shCtl (n = 7) or sh*Cdk6* (n = 9). Unpaired *t* test. (G) Representative fluorescent images of TA muscle from KO/*mdx* mice injected with AAV9 shCtl or sh*Cdk6* stained with MyoD, Pax7, and DAPI. White and yellow arrows indicate Pax7$^+$/MyoD$^+$, and Pax7$^+$/MyoD$^-$ cells, respectively. Scale bar: 50 μm. (H) Number of Pax7$^-$/MyoD$^+$, Pax7$^+$/MyoD$^+$, and Pax7$^+$/MyoD$^-$ cells in TA muscle of KO/*mdx* mice injected with AAV9 shCtl or sh*Cdk6*. Three fields of views/mouse (n = 5/group). Unpaired *t* test. Data are presented as the mean ± SEM. *$P < 0.05$, **$P < 0.01$, ****$P < 0.0001$. Source data are available online for this figure.

developed AMOs against miR-33a/b to inhibit their function in vivo and demonstrated that local and systemic administration of these oligonucleotides improved the dystrophic muscle phenotype in *mdx* mice, indicating potential clinical applications. Notably, systemic administration of anti-miR-33b ameliorated declined exercise capacity, as measured using a treadmill endurance test. Furthermore, miR-33b inhibition upregulated these miR-33 target genes in myotubes from human iPS cells derived from a patient with DMD. These findings indicate that miR-33a/b reduce muscle regeneration capacity by suppressing SC expansion and that miR-33a/b inhibition may be a novel therapeutic approach for patients with muscular dystrophy, including DMD.

miR-33a expression was upregulated during the differentiation of C2C12 cells or mouse primary myoblasts and muscle regeneration following CTX injury. Furthermore, the muscles of *mdx* mice showed elevated expression of miR-33a. These findings suggest that miR-33a influences muscle differentiation and regeneration. Our experiments revealed that miR-33a-KO accelerated muscle regeneration in response to CTX injury and ameliorated the dystrophic phenotype in *mdx* mice. In contrast, as a gain-of-function, miR-33b-KI mice deteriorated to the dystrophic phenotype in *mdx* mice. Therefore, miR-33 is a negative regulator of muscle regeneration, and this assertion is supported by a previous report on duck myoblasts (Li et al, 2020).

SCs are located beneath the basal lamina of the muscle fibers and function as the stem cells of skeletal muscle. Quiescent SCs undergo activation in response to muscle injury and proliferate to generate myogenic progenitor cells (myoblasts) that contribute to the regeneration and repair of injured muscle. These cells are indispensable for muscle regeneration, and impaired expansion has been observed in the muscles of *mdx* mice and patients with DMD (Tedesco et al, 2010). Asymmetric cell division, which contributes to myoblast production and is important for regeneration, is impaired in dystrophin-deficient SCs (Dumont et al, 2015; Tedesco et al, 2010). *Pax7* is a specific marker expressed in SCs and is indispensable for normal functions, including expansion (von Maltzahn et al, 2013), which is suppressed during myogenic differentiation. In response to CTX injury, the number of Pax7$^+$ cells increased more rapidly in the muscles of miR-33a-KO mice than in the control mice. Similarly, an increased number of Pax7$^+$ positive cells was observed in miR-33a-KO/*mdx* mice, whereas the number of Pax7$^+$ cells was decreased in miR-33b-KI/*mdx* mice. These observations were also confirmed by primary myoblast proliferation in vitro.

*Cdk6* expression was increased in miR-33a-KO/*mdx* mice and suppressed in miR-33b-KI/*mdx* mice and was also increased following treatment with AMOs against miR-33a/b. *Cdk6* is crucial

for accelerating cell proliferation by phosphorylating Rb and related proteins in the G1 phase of the cell cycle (Goel et al, 2022). We observed increased levels of phosphorylated Rb and PCNA in the TA muscle of miR-33a-KO/*mdx* mice. AAV9-mediated *Cdk6* knockdown reversed the increase in the number of SCs in the muscle of miR-33a-KO/*mdx* mice and the increased proliferation rate of primary myoblasts. These findings indicate that miR-33a/b suppress SC proliferation by targeting *Cdk6*, leading to reduced muscle regeneration and increased fibrosis. *Cdk6* knockdown reversed all beneficial muscle phenotypes in miR-33a-KO/*mdx* mice, indicating that *Cdk6* upregulation primarily contributes to these phenotypes in miR-33a-KO/*mdx* mice.

FST inhibits the function of other members of the TGF-β superfamily, including myostatin, a negative regulator of muscle development and growth that induces muscle atrophy-related genes, such as *MuRF1* and *Atrogin1*, through the Smad signaling pathway (Baig et al, 2022). Myostatin also negatively regulates SC activation and self-renewal (McCroskery et al, 2003), partly through Pax7-inhibition via Erk1/2 signaling (McFarlane et al, 2008). In contrast, *FST* promotes muscle growth through SC proliferation (Gilson et al, 2009), muscle healing and regeneration after a laceration injury (Zhu et al, 2011), and regeneration to repair various muscle injuries, such as CTX-induced injury, hind-limb immobilization, and ovariectomy (Yaden et al, 2014). *FST* also increases muscle mass and improves dystrophic pathology in *mdx* mice (Benabdallah et al, 2008; Nakatani et al, 2008). Although previously observed in duck myoblasts (Li et al, 2020), we demonstrated that *Fst* was a target gene of miR-33 both in vitro and in vivo. This relationship is highly conserved among species, including humans. FST was increased in miR-33a-KO/*mdx* mice and suppressed in miR-33b-KI/*mdx* mice. AAV9-mediated shRNA targeting *Fst* reversed the phenotype of miR-33a-KO/*mdx* mice in a similar manner to that of *Cdk6*. Moreover, miR-33a/b inhibition increased muscle expression of FST. Therefore, we consider *FST* as another significant regulator of muscle regeneration in addition to *CDK6*, targeted by miR-33a/b.

We investigated *Abca1* as an additional miR-33 target gene for these phenotypes. In vivo, *Abca1* knockdown exacerbated fibrosis in miR-33a-KO/*mdx* mice without altering myogenic markers, including *Pax7*. ABCA1 attenuates inflammation by modulating the quantity of lipid rafts (Bi et al, 2015). Consequently, *Abca1* upregulation in muscle may contribute to the amelioration of fibrosis through alternative pathways distinct from an increase in stem cells. Furthermore, systemic AMO treatment or global miR-33a deficiency upregulates hepatic ABCA1, resulting in elevated serum HDL-C levels (Horie et al, 2010a; Najafi-Shoushtari et al, 2010; Rayner et al, 2010) (Table 1). Since HDL-C possesses anti-

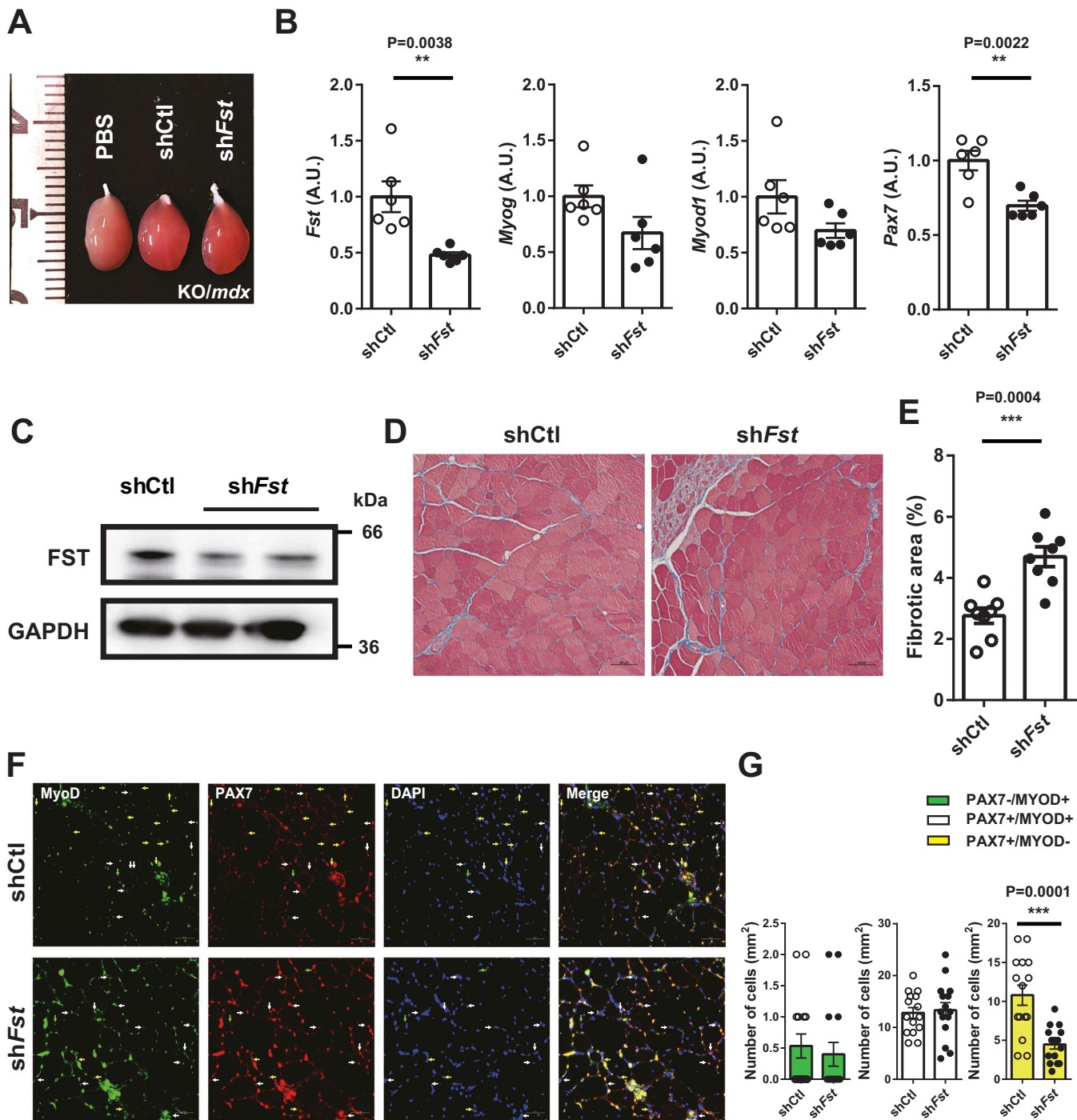

**Figure 6. *Fst* knockdown reverses the beneficial phenotype of miR-33a-KO/*mdx* mice, similar to *Cdk6*.**

(A) Representative images of TA muscle from KO/*mdx* mice injected with PBS, AAV9 shRNA control (shCtl), or AAV9 shRNA against *Fst* (sh*Fst*). (B) Expression of *Fst*, *Myog*, *Myod1*, and *Pax7* in the TA muscle of KO/*mdx* mice injected with AAV9 shCtl or sh*Fst* (n = 6/group). Unpaired *t* test. (C) Western blotting for FST and GAPDH in the TA muscle of KO/*mdx* mice injected with AAV9 shCtl or sh*Fst*. (D) Representative images of Masson trichrome staining of TA muscle of KO/*mdx* mice injected with AAV9 shCtl or sh*Fst*. Scale bar: 100 μm. (E) Percentage of fibrotic area in the TA muscle of KO/*mdx* mice injected with AAV9 shCtl or sh*Fst* (n = 8/group). Unpaired *t* test. (F) Representative fluorescent images of TA muscle from KO/*mdx* mice injected with AAV9 shCtl or sh*Fst* and stained with MyoD, Pax7, and DAPI. Green, white and yellow arrows indicate Pax7$^-$/MyoD$^+$, Pax7$^+$/MyoD$^+$, and Pax7$^+$/MyoD$^-$ cells, respectively. Scale bar: 50 μm. (G) Number of Pax7$^-$/MyoD$^+$, Pax7$^+$/MyoD$^+$, and Pax7$^+$/MyoD$^-$ cells in TA muscle of KO/*mdx* mice injected with AAV9 shCtl or sh*Fst*. Three fields of view/mouse (n = 5/group). Unpaired *t* test. Data are presented as the mean ± SEM. **$P < 0.01$, ***$P < 0.001$. Source data are available online for this figure.

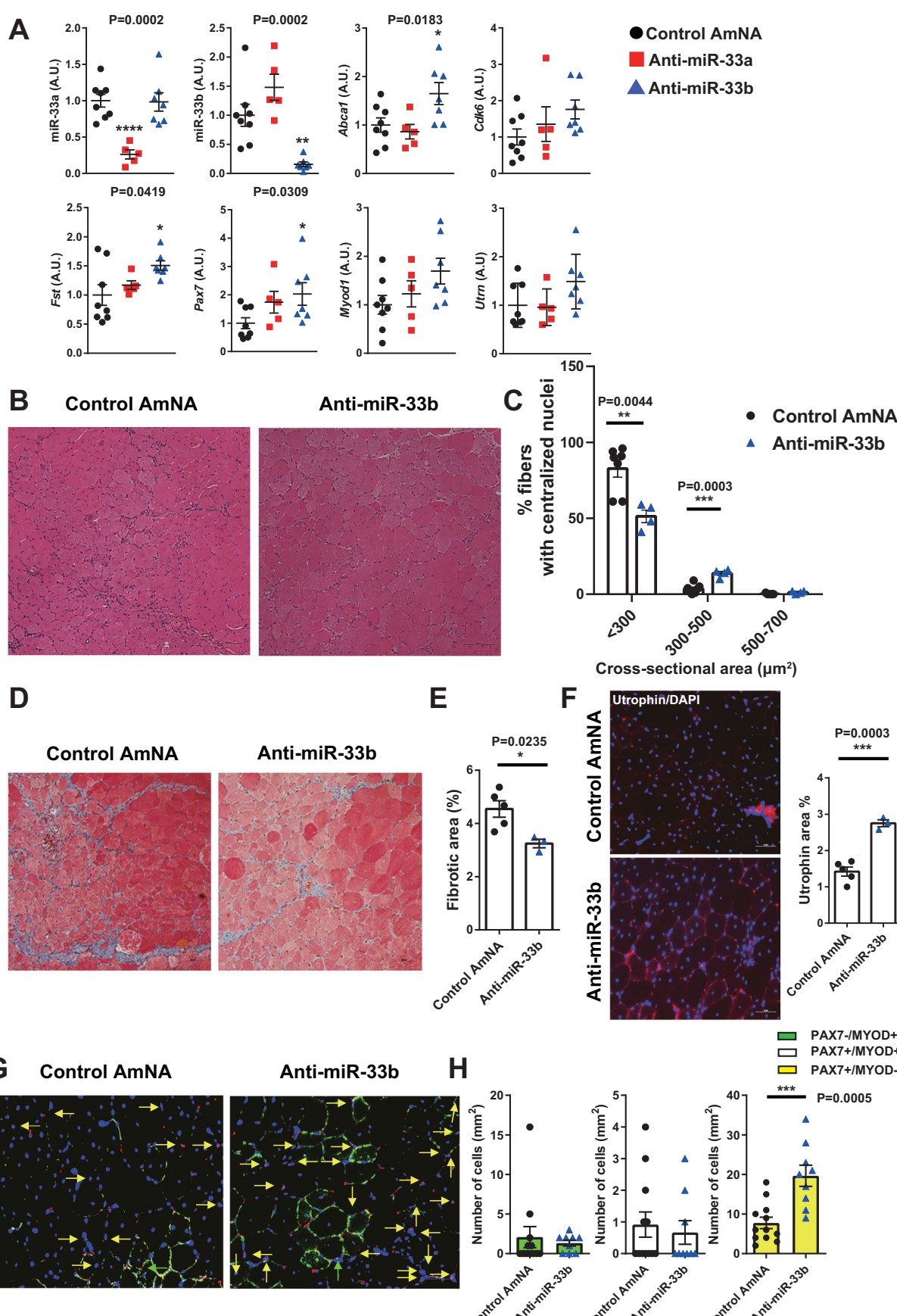

**Figure 7. Local delivery of AMO-33b ameliorates the dystrophic phenotype in the muscles of miR-33b-KI/*mdx* mice.**

(A) miR-33a, miR-33b, *Abca1*, *Cdk6*, *Fst*, *Pax7*, *Myod1*, and *Utrn* expression in the TA muscle of miR-33b-KI/*mdx* mice injected with control AmNA, anti-miR-33a, or anti-miR-33b ($n = 5$–8/group). One-way ANOVA with Tukey post-hoc test. (B) Representative images of HE staining of TA muscle of miR-33b-KI/*mdx* mice injected with control AmNA or anti-miR-33b. Scale bar: 100 μm. (C) Size distribution of muscle fibers in TA muscle of miR-33b-KI/*mdx* mice injected with control AmNA ($n = 7$) or anti-miR-33b ($n = 4$). Unpaired *t* test. (D) Representative images of Masson trichrome staining of TA muscle of miR-33b-KI/*mdx* mice injected with control AmNA or anti-miR-33b. Scale bar: 100 μm. (E) Percentage of fibrotic area in the TA muscle of miR-33b-KI/*mdx* mice injected with control AmNA ($n = 5$) or anti-miR-33b ($n = 3$). Unpaired *t* test. (F, left) Representative fluorescent images of TA muscle from miR-33b-KI/*mdx* mice injected with control AmNA or anti-miR-33b and stained with utrophin and DAPI. Scale bar: 50 μm. (F, right) Quantification of the utrophin-positive area in TA muscle from miR-33b-KI/*mdx* mice injected with control AmNA ($n = 5$) or anti-miR-33b ($n = 3$). Unpaired *t* test. (G) Representative fluorescent images of TA muscle of miR-33b-KI/*mdx* mice injected with control AmNA or anti-miR-33b and stained with MyoD, Pax7, and DAPI. Green and yellow arrows indicate Pax7$^-$/MyoD$^+$ and Pax7$^+$/MyoD$^+$ cells, respectively. Scale bar: 50 μm. (H) Number of Pax7$^-$/MyoD$^+$, Pax7$^+$/MyoD$^+$, and Pax7$^+$/MyoD$^-$ cells in the TA muscle of miR-33b-KI/*mdx* mice injected with control AmNA ($n = 4$) or anti-miR-33b ($n = 3$). Three fields of view/mice. Unpaired *t* test. Data are presented as the mean ± SEM. *$P < 0.05$, **$P < 0.01$, ***$P < 0.001$. Source data are available online for this figure.

inflammatory properties (Rosenson et al, 2016), this suggests that it exerts additional beneficial effects on muscle.

Muscular dystrophy is a class of diseases, among which DMD is a severe, progressive disorder caused by mutations in *DMD* that result in the absence of functional dystrophin protein (Duan et al, 2021). Dystrophin is the primary component of the dystrophin-associated protein complex (DAPC) at the sarcolemma, which stabilizes the muscle cell membrane. DMD pathology is caused by a lack of functional dystrophin. Thus, restoring dystrophin function or expression is a potential therapeutic approach for patients with DMD (Markati et al, 2022). Although no curative treatment for DMD currently exists, recent technological advances have led to the development of novel therapies for specific subsets of patients with DMD. Exon 51 skipping therapy has been approved by the U.S. Food and Drug Administration, but it is only effective for patients with mutations in exon 51 of *DMD* (Echevarría et al, 2018; McDonald et al, 2021). Similarly, stop codon read-through can be applied to certain subgroups of patients (McDonald et al, 2017; Welch et al, 2007). In contrast, miR-33 inhibition enhances muscle regeneration capacity by increasing the number of SCs via *Cdk6* and *Fst* upregulation. Moreover, *Abca1* upregulation contributes to fibrosis reduction by suppressing inflammation. Therefore, this approach can be applied to all categories of muscular dystrophy, including DMD and BMD.

Utrophin is a structural and functional dystrophin paralog that shares 80% homology with dystrophin and exhibits functional redundancy (Guiraud and Davies, 2017; Szwec et al, 2024). During the natural repair process in *mdx* mice, utrophin levels were reported to increase by 1.8-fold compared with normal levels (Guiraud and Davies, 2017). Depending on the quantity of utrophin, dystrophic pathology is mitigated in truncated or full-length utrophin transgenic mice (Tinsley et al, 1998; Tinsley et al, 1996). This may be attributed to the formation of the utrophin-associated protein complex (UAPC), instead of DAPC, thereby enhancing sarcolemma stability. Thus, modulation of utrophin levels may be an alternative therapeutic approach for DMD patients (Guiraud and Davies, 2017; Soblechero-Martín et al, 2021; Szwec et al, 2024). miR-33a-KO/*mdx* mice showed enhanced utrophin expression, while miR-33b-KI/*mdx* mice showed reduced utrophin expression, and the inhibition of miR-33a/b restored utrophin expression levels. Therefore, miR-33a/b inhibition exerts a beneficial effect on injured muscles from the perspective of utrophin levels and leads to improved muscle function, potentially reflecting differences in muscle regeneration and maturation through altered expression of miR-33a/b. β2-syntrophin (*SNTB2*) is a component of DAPC and UAPC (Froehner et al, 1997; Soblechero-Martín et al, 2021;

Szwec et al, 2024) and the C-terminal of ABCA1 interacts with the β2-syntrophin/utrophin complex (Buechler et al, 2002). In addition to *ABCA1*, we found that *SNTB2* also contains a potential miR-33 binding site in its 3′-UTR, as confirmed by a 3′-UTR luciferase assay (Appendix Fig. S11G). Therefore, miR-33a/b may influence utrophin levels by destabilizing UAPC through targeting *ABCA1* and *SNTB2*, in addition to reflecting muscle regeneration and maturation. Further investigation is necessary to elucidate the precise mechanisms underlying alterations in utrophin levels.

Antisense oligonucleotides are potent tools for gene modulation, and their application represents a promising era for novel therapies. Several modifications have been developed to enhance their efficiency. In this study, we used AMOs modified with AmNA and phosphonothioate to augment their binding ability and stability (Miyagawa et al, 2023; Yamasaki et al, 2022). AmNA, an analog of locked nucleic acid (LNA) with modification of the amide bond bridged between the 2′ and 4′ carbons of ribose, demonstrates higher knockdown efficiency and safety than natural antisense oligonucleotides and LNAs (Yahara et al, 2012; Yamamoto et al, 2015). We directly administered AMOs into the muscle and confirmed the beneficial effects using efficient knockdown in *mdx* mice. Furthermore, the systemic administration of AMOs effectively ameliorated the dystrophic phenotype, including exercise capacity, without apparent adverse effects. RNA sequence analysis of muscle following anti-miR-33b administration revealed not only changes in miR-33 target genes but also widespread alterations in genes associated with cell cycle and muscle regeneration. Experiments on myotubes derived from human iPS cells of a patient with DMD showed that anti-miR-33b treatment increased the expressions of miR-33 target genes and utrophin, consistent with the observations in mice. These findings suggest that miR-33 inhibition may possess therapeutic potential for patients with DMD. For clinical application of AMOs, it is essential to address organ selectivity and enhance safety, stability, and efficacy through additional research.

In conclusion, this study elucidates the critical role of miR-33a/b in muscle regeneration. miR-33a/b inhibit SC proliferation and exacerbate muscle regeneration and fibrosis by targeting *Cdk6*, *Fst*, and *Abca1*. miR-33 inhibition ameliorated the dystrophic phenotype in *mdx* mice by upregulating these target genes. These genes were also upregulated by miR-33 inhibition in myotubes differentiated from human iPS cells of a DMD patient. Thus, miR-33a/b inhibition may be a novel therapeutic approach for patients with muscular dystrophy, including DMD, by targeting multiple pathways.

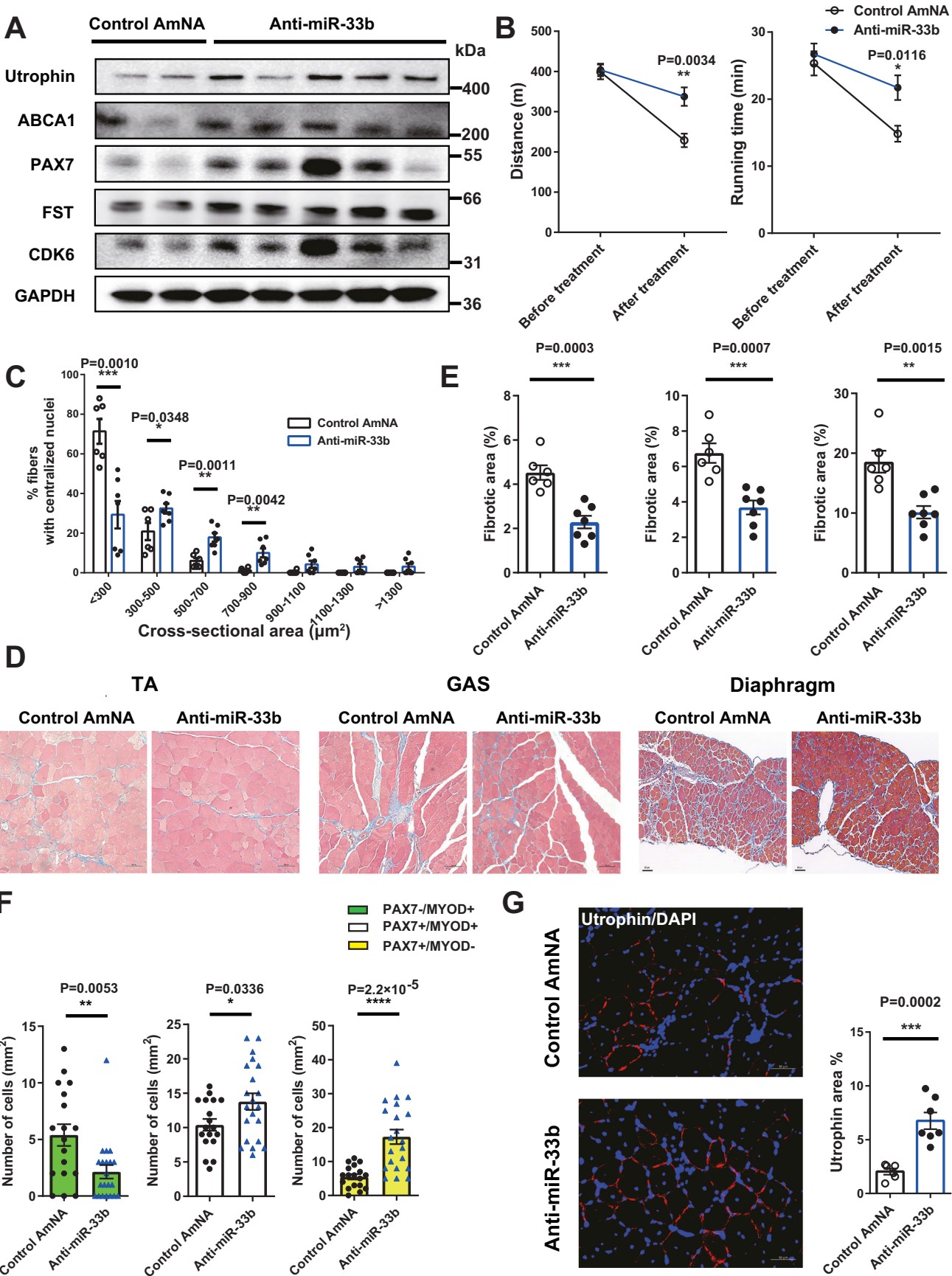

**Figure 8. Systemic administration of anti-miR-33b ameliorates the dystrophic phenotype and exercise capacity in miR-33b-KI/*mdx* mice.**

(A) Western blotting for utrophin, ABCA1, PAX7, FST, CDK6, and GAPDH in the TA muscle of miR-33b-KI/*mdx* mice treated with control AmNA or anti-miR-33b. (B) Changes in running distance and running time during the treadmill endurance test in miR-33b-KI/*mdx* mice treated with control AmNA ($n = 6$) or anti-miR-33b ($n = 7$). Unpaired $t$ test. (C) Size distribution of muscle fibers in the TA muscle of miR-33b-KI/*mdx* mice treated with control AmNA ($n = 6$) or anti-miR-33b ($n = 7$). Unpaired $t$ test. (D) Representative images of Masson trichrome staining of TA muscle, GAS muscle, and diaphragm of miR-33b-KI/*mdx* mice treated with control AmNA or anti-miR-33b. Scale bars: 100 μm (left and middle) and 50 μm (right). (E) Percentage of fibrotic area in the TA muscle, GAS muscle, and diaphragm of miR-33b-KI/*mdx* mice treated with control AmNA ($n = 6$) or anti-miR-33b ($n = 7$). Unpaired $t$ test. (F) Number of Pax7⁻/MyoD⁺, Pax7⁺/MyoD⁺, and Pax7⁺/MyoD⁻ cells in the TA muscle of miR-33b-KI/*mdx* mice injected with control AmNA ($n = 6$) or anti-miR-33b ($n = 7$). Three fields of view/mice. (G, left) Representative fluorescent images of TA muscle from miR-33b-KI/*mdx* mice treated with control AmNA or anti-miR-33b and stained with utrophin and DAPI. Scale bar: 50 μm. (G, right) Quantification of the utrophin-positive area of TA muscle from miR-33b-KI/*mdx* mice treated with control AmNA ($n = 6$) or anti-miR-33b ($n = 7$). Unpaired $t$ test. Data are presented as the mean ± SEM. *$P < 0.05$, **$P < 0.01$, ***$P < 0.001$. Source data are available online for this figure.

**Table 1. Serum data of mice administered AMOs at a dose of 20 mg/kg bw.**

| Serum analyte | Control AmNA | Anti-miR-33b | P value |
|---|---|---|---|
| AST (IU/L) | 1098 ± 312 | 634 ± 295 | *$P = 0.0188$ |
| ALT (IU/L) | 151 ± 38 | 105 ± 34 | *$P = 0.0424$ |
| LDH (IU/L) | 4706 ± 1354 | 2301 ± 787 | **$P = 0.0021$ |
| CK (IU/L) | 5769 ± 3798 | 2935 ± 732 | $P = 0.0775$ |
| T-CHO (mg/dL) | 72 ± 7.8 | 103 ± 6.7 | ****$P = 1.78 \times 10^{-6}$ |
| HDL-C (mg/dL) | 38 ± 2.1 | 54 ± 5.4 | ****$P = 6.77 \times 10^{-6}$ |

*AMO* anti-microRNA oligonucleotide, *AmNA* amido-bridged nucleic acid, *ALT* alanine aminotransferase, *AST* aspartate aminotransferase, *bw* body weight, *CK* creatinine kinase, *HDL-C* high-density lipoprotein cholesterol, *LDH* lactate dehydrogenase, *T-CHO* total cholesterol.

Serum was obtained from miR-33b-KI/*mdx* mice systemically administered 20 mg/kg bw of control AmNA ($n = 6$) or anti-miR-33b ($n = 7$) weekly for 4 weeks. Unpaired $t$ test. *$P < 0.05$, **$P < 0.01$, ****$P < 0.0001$.

# Methods

### Reagents and tools table

| Reagent/resource | Reference or source | Identifier or catalog number |
|---|---|---|
| **Experimental models** | | |
| *mdx* mice | CLEA Japan | B10-*mdx* (C57BL/10ScSn-Dmd^mdx^/JicJcl) |
| miR-33a-KO mice | Proc Natl Acad Sci U S A.2010 Oct 5;107(40):17321-6. https://doi.org/10.1073/pnas.1008499107 | Originally generated |
| miR-33b-KI mice | Sci Rep.2014 Jun 16:4:5312. https://doi.org/10.1038/srep05312. | Originally generated |
| **Recombinant DNA** | | |
| psiCHECK-2 | Promega | C8021 |
| **Antibodies** | | |
| Anti-CDK6 | CST | #3136 |
| Anti-ABCA1 | Novus | NB400-105 |
| Anti-PAX7 | DSHB | DSHB-PAX7-SA-5 |
| Anti-Utrophin | Santa cruz | SC-33700 (8 A4) |
| Anti-MyoD | Santa cruz | SC-32758 (5.8A) |
| Anti-Myogenin | Santa cruz | SC-12732 (F5D) |
| Anti-MHC | Santa cruz | SC-32732 (F59) |
| Anti-PCNA | Santa cruz | SC-56 (PC10) |

| Reagent/resource | Reference or source | Identifier or catalog number |
|---|---|---|
| Anti-Ki67 | Abcam | Ab16667 (SP6) |
| Anti-Phospho-Rb | CST | #8516 |
| Anti-Rb | BD Bioscience | 554136 |
| Anti-FST | Santa cruz | SC-365003 (C-8) |
| Anti-Cyclin D1 | Abcam | ab92566 |
| Anti-GAPDH | CST | #2118 |
| DAPI solution | DOJINDO | D523 |
| Alexa Fluor 594 donkey anti-mouse IgG (H + L) | ThermoFisher | A21203 |
| Alexa Fluor 488 donkey anti-rabbit IgG (H + L) | ThermoFisher | A21206 |
| AffinPure Fab Fragment rabbit anti-mouse IgG | Jackson | 315-007-003 |
| **Oligonucleotides and other sequence-based reagents** | | |
| PCR primers for mRNA | SIGMA | Listed in Appendix Table 3 |
| PCR primers for microRNA | ThermoFisher | TaqMan™ MicroRNA Assay: hsa-miR-33a-5p hsa-miR-33b U6 snRNA: |
| **Chemicals, enzymes and other reagents** | | |
| DMEM | Nacalai Tesque | 08458-16 |
| Ham's F-10 Medium | Gibco-Invitrogen | 11550-043 |
| Penicillin/Streptomycin | Gibco-Invitrogen | 15070-063 |
| Fetal Bovine Serum | Sigma-Aldrich | 172012-500 ML |
| Horse Serum | Gibco-Invitrogen | 16050122 |
| Collagen (from calf skin) | Sigma-Aldrich | C8919-20ML |
| bFGF, human, Recombinant | Wako | 060-04543 |
| Collagenase Type2 | Worthington | CLS-2 |
| Cell Strainer (70μm) | BD Biosciences | 352350 |
| Cardiotoxin | Sigma-Aldrich | C9759-1MG |
| 29 G Insulin Syringe | TERUMO | SS-05M2913 |
| Bovine Serum Albumin | Sigma-Aldrich | A7030-100MG |
| Lectin from Tritium Vulgaris | Sigma-Aldrich | L4895-5MG |

| Reagent/resource | Reference or source | Identifier or catalog number |
|---|---|---|
| MTT | DOJINDO | 341-01823 |
| Cell Tray | Sumitomo Bakelite | MS-12400 |
| AAV-293 Cells | Agilent | #240073 |
| Proteinase K | Sigma-Aldrich | P2308 |
| Benzonase | Novagen | 70746-4 |
| PicaGENE Dual Sea Pansy Luminescence Kit | TOYOINK GROUP | PGD-S |
| Verso cDNA Synthesis Kit | ThemoFisher | AB1453B |
| TaqMan MicroRNA Reverse Transcription Kit | Applied Biosystems | 4366597 |
| TaqMan Universal Master Mix II, no UNG | Applied Biosystems | 4440040 |
| THUNDERBIRD SYBER qPCR Mix | TOYOBO | QPS-201 |
| VECTASHIELD Mounting Medium | VECTOR LABORATORIES | H-1000 |
| Stemfit AK02N | Takara Bio Inc | AJ100 |
| PECM | Reprocell | RCHEMD001 |
| Matrigel | Falcon | #356231 |
| Accutase | Nacalai Tesque | #12679-54 |
| Y-27632 | Nacalai Tesque | #0894584 |
| Doxycyclin Hyclate | LKT Labs | #D5897 |
| αMEM | Nacalai Tesque | #2144405 |
| KSR | Invitrogen | #10828028 |
| DMEM (High glucose) | Fujifilm | 048-29763 |
| IGF-1 | Peprotech | #100-11 |
| SB431542 | Wako | #198-16543 |
| 2-mercaptoethanol (2-ME) | Nacalai Tesque | #2143882 |
| Puromycin | Nacalai Tesque | 19752-64 |
| Easy iMatrix-511 silk | Takara Bio Inc | T313 |
| RNAiMAX | ThermoFisher | 13778150 |
| **Software** | | |
| ImageJ | NIH | ImageJ 1.44p |
| GraphPad Prism | GraphPad Software, Inc. | GraphPad Prism 5.04 and 6.05 |
| **Other** | | |
| NovaSeqX | Macrogen, Inc | |

## Mice and animal care

C57BL/10-*mdx* mice were purchased from CLEA Japan, Inc. miR-33a-KO and miR-33b-KI C57BL/6J mice were generated as previously described (Horie et al, 2014b; Horie et al, 2010a) miR-33a-KO and miR-33b-KI mice were crossed with *mdx* mice to produce miR-33a-KO/*mdx* and miR-33b-KI/*mdx* mice with a mixed C57BL/6 J and C57BL/10 background. Littermates were used as the controls. All experiments were performed using male mice. The mice were maintained in temperature-controlled rooms with a

14:10 h light/dark cycle under specific pathogen-free conditions at the Institute of Laboratory Animals, Kyoto University Graduate School of Medicine. The study protocol was approved by the Kyoto University Ethics Review Board.

## Primary myoblast isolation

Primary myoblasts were isolated from the TA muscles of 7–8-week-old male mice (Danoviz and Yablonka-Reuveni, 2012; Motohashi et al, 2014). The muscles were minced, digested in 0.2% type II collagenase solution (10% fetal bovine serum [FBS] in Dulbecco modified Eagle medium [DMEM]) at 37 °C for 60 min and homogenized using an 18 G needle. Then the homogenate was incubated at 37 °C for 15 min, diluted to 50 mL with 2% FBS in DMEM, and mixed to create a single-cell suspension. After filtration through a 70-μm cell strainer and centrifugation (2000 rpm, 4 °C, 5 min), the supernatant was discarded. The cells were washed in 10 mL of 2% FBS in DMEM and cultured in growth medium (GM; F-10 Ham medium with 20% FBS, 2 ng/mL basic fibroblast growth factor, and 1% penicillin/streptomycin) on collagen-coated plates at 37 °C under 5% $CO_2$. Differentiation was induced in a differentiation medium (DM) containing 2% horse serum, and the cells were cultured for 24–72 h.

## Muscle injury and regeneration

Muscle regeneration was induced by CTX injection (Guardiola et al, 2017). Briefly, 8-week-old mice were anesthetized and injected with 50 μl of 10 μM CTX in PBS into the right TA muscle and PBS alone into the left TA muscle (negative control). The muscle tissues were harvested after 3, 7, 14, and 21 days to evaluate regeneration and repair.

## Western blot analysis

The muscle tissue samples were homogenized in chilled lysis buffer (50 mM Tris-HCl [pH 7.4], 75 mM NaCl, 1% Triton X-100, complete Mini Protease Inhibitor Cocktail [Roche], 0.5 mM NaF, and $Na_3VO_4$) using a polytron homogenizer. Then the protein samples (15 μg) were separated using NuPAGE Bis-Tris Mini Protein Gels, 4–12% (Invitrogen), and transferred to nitrocellulose blotting membranes per the manufacturer's instructions (Horie et al, 2010b). Utrophin, a high-molecular-weight protein, was electrophoresed for 4 h at 80 mA and transferred to a nitrocellulose membrane overnight (50 V at 4 °C) (Han et al, 2016). The membranes were blocked with PBS containing 5% nonfat milk for 30 min and incubated overnight at 4 °C with the following primary antibodies: anti-PAX7 (Developmental Studies Hybridoma Bank; 1:500), anti-CDK6 (Cell Signaling Technology; 1:500), anti-ABCA1 (Novus Biologicals; 1:500), anti-MyoG (Santa Cruz Biotechnology; 1:500), anti-MyoD (Santa Cruz Biotechnology; 1:500), anti-utrophin (Santa Cruz Biotechnology; 1:500), anti-PCNA (Santa Cruz Biotechnology; 1:500), anti-FST (Santa Cruz Biotechnology; 1:150), anti-phospho-Rb (Santa Cruz Biotechnology; 1:500), anti-Rb (BD Biosciences; 1:500), and anti-GAPDH (Cell Signaling Technology; 1:3000). After washing with PBS with 0.05% Tween-20 (PBS-T), the membranes were incubated with a secondary antibody for 1 h at room temperature and washed again with PBS-T. Labeled proteins were detected using Pierce ECL Plus Western Blotting Substrate (ThermoFisher Scientific) or Pierce ECL Western Blotting

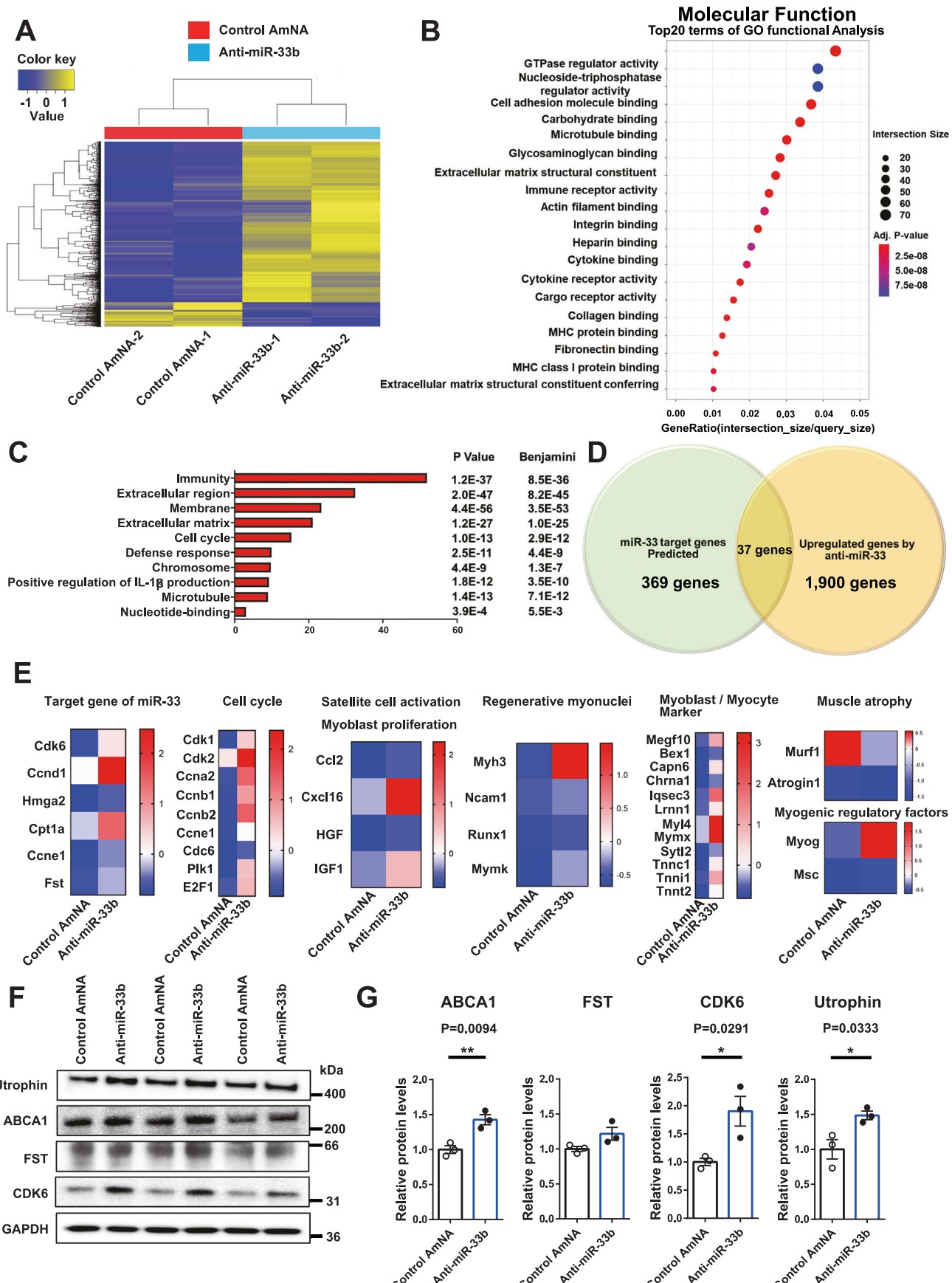

**Figure 9. RNA sequence analysis of muscle following systemic administration of anti-miR-33b and analysis of myotubes from DMD patient-derived iPS cells with anti-miR-33b.**

(A) Heatmap of the hierarchical clustering of DEGs in muscles following systemic administration of control AmNA or anti-miR-33b ($n = 2$/group). (B) Gene ontology terms related to molecular function. Differential expression analysis was performed using likelihood ratio tests and quasi-likelihood F-tests in the edgeR package, and $P$ values were adjusted using the Benjamini–Hochberg method. Gene ontology functional analysis was conducted using hypergeometric tests. (C) Pathway analysis for the DEGs. Pathway analysis was performed using DAVID (Database for Annotation, Visualization, and Integrated Discovery). Statistical significance was evaluated using modified Fisher's exact tests (EASE score), and $P$ values were adjusted using the Benjamini–Hochberg method. (D) Venn diagram analysis between miR-33 target prediction and upregulated genes following anti-miR-33b treatment. (E) Representative DEGs following anti-miR-33b treatment categorized by the indicated molecular functions. (F) Western blotting for utrophin, ABCA1, FST, CDK6, and GAPDH in myotubes differentiated from human iPS cells of a patient with DMD (CiRA00111) treated with control-AmNA or anti-miR-33b ($n = 3$/group). (G) Densitometric analysis of ABCA1, FST, CDK6 and utrophin in myotubes differentiated from human iPS cells of a patient with DMD (CiRA00111) treated with control-AmNA or anti-miR-33b ($n = 3$/group). Unpaired $t$ test. Data are presented as the mean ± SEM. *$P < 0.05$, **$P < 0.01$. Source data are available online for this figure.

Substrate (ThermoFisher Scientific) with a LAS-4000 Mini system (Fujifilm). The proteins were quantified densitometrically using ImageJ 1.44p software (National Institutes of Health).

## Histology

Mice were administered an isoflurane overdose followed by perfusion with 4% paraformaldehyde (PFA). The muscle was excised and fixed overnight at 4 °C in 4% PFA. The next day, the tissue was dehydrated in 70% ethanol and embedded in paraffin blocks. Sectioned samples were deparaffinized and stained with HE and Masson trichrome stain. Images were captured using an Axio Observer microscope (Zeiss) and analyzed using ImageJ 1.44p software. For quantitative analysis, the cross-sectional area of approximately 100 myofibers was measured in HE-stained TA muscle sections (Wu et al, 2015) to calculate the fiber distribution. Areas of fibrosis were assessed using Masson trichrome-stained TA muscle (Liu et al, 2012).

## CK assay

Blood was collected from the inferior vena cava of 8-week-old anesthetized mice. Following centrifugation at 4 °C, the serum was decanted and stored at −80 °C for CK measurement on a Hitachi 7180 auto analyzer (Oriental Yeast Co., Ltd., Nagahama, Japan) using standard methods.

## RNA extraction and quantitative reverse transcription polymerase chain reaction (qRT-PCR)

Total RNA was extracted using TriPure Isolation Reagent (Roche). Then 1 µg of RNA was transcribed into cDNA using a Verso cDNA Synthesis Kit (ThermoFisher Scientific) per the manufacturer's protocol. qRT-PCR was performed using THUNDERBIRD SYBR qPCR Mix (TOYOBO) on a StepOnePlus Real-Time PCR System (Applied Biosystems) for 40 amplification cycles per the manufacturer's protocol. Gene expression was normalized against GAPDH. The primers are detailed in Appendix Table S3.

## Quantitative PCR (qPCR) for miRNAs

Total RNA was extracted using TriPure Isolation Reagent. miR-33a/b levels were quantified using TaqMan MicroRNA assays (Applied Biosystems) and analyzed using an ABI StepOnePlus Real-time PCR System. The samples were normalized against U6 snRNA expression levels.

## Lentivirus production and DNA transduction

Lentiviral stocks were generated from 293FT cells according to the manufacturer's protocol (Invitrogen) (Horie et al, 2010b). Briefly, the virus-containing medium was collected at 48 h post-transfection and passed through a 0.45-µm filter. A single round of lentiviral infection was performed by replacing the medium with virus-containing medium (8 µg/mL polybrene), followed by centrifugation (2500 rpm, 30 min, 32 °C). The cells were analyzed 3 days after DNA transduction.

## Immunohistochemistry

Primary myoblasts were seeded in six-well plates ($3 \times 10^5$ cells/cm² per well) containing GM. When the cells reached 70%–80% confluency, the medium was replaced with DM for 24–72 h. The cells were fixed with 4% PFA for 10 min and permeabilized in 0.1% Triton X-100 in PBS for 10 min at room temperature. After blocking with 5% donkey serum for 30 min, the cells were incubated with primary antibodies (myosin heavy chain [MHC], 1:50) for 1 h, followed by Alexa Fluor 594-conjugated secondary antibodies (ThermoFisher Scientific; 1:200) with DAPI (1 µg/mL) for 1 h at room temperature. The slides were washed, mounted in Vectashield mounting medium (Vector Laboratories), and examined using an Axio Observer microscope. The fusion index was determined by counting the total number of nuclei and multinucleated myotube nuclei in each field of view using ImageJ 1.44p software and calculating their ratio (Liu et al, 2012).

Paraffin-embedded muscle tissue sections underwent immunostaining. Briefly, the sections were deparaffinized, rinsed in PBS, and autoclaved for heat-mediated antigen retrieval in EDTA buffer (pH 8.0) for Ki-67 staining and in citrate buffer (pH 6.0) for PAX7, MyoD, utrophin, and lectin staining (Hindi and Kumar, 2016; Podkalicka et al, 2020). The sections were blocked with 5% donkey serum in PBS at room temperature for 15 min and incubated overnight at 4 °C with primary antibodies. Fab fragments were used for double staining. The slides were rinsed three times and incubated with Alexa Fluor 488- or 594-conjugated secondary antibodies with DAPI (1 µg/mL) at room temperature for 1 h. After washing, the slides were mounted in Vectashield mounting medium for microscopic examination. Three random images per skeletal muscle were obtained from each mouse to quantify Ki-67-, Pax7-, and DAPI-positive cells, and Pax7-negative or positive, MyoD-negative or positive, and DAPI-positive cells (i.e., Ki-67[+]/Pax7[+], Pax7[−]/MyoD[+], Pax7[+]/MyoD[+], and Pax7[+]/MyoD[−]). The

images were captured using an Axio Observer microscope (Zeiss) and analyzed using ImageJ 1.44p software. To ensure consistency across the dataset, identical brightness and contrast parameters were applied to all images.

## MTT assay

Primary myoblast proliferation was assessed using an MTT assay. Briefly, cells were seeded in six-well culture dishes ($2.5 \times 10^4$ cells/well) and incubated in 2 mL of medium containing 10% FBS. Then, 200 µL MTT solution (5 mg/mL) was added at specified time points. The cells were incubated at 37 °C for 4 h and lysed with the addition of 2 mL of acidified isopropanol (40 mmol/L HCl). Absorbance was measured at 595 nm using an ARVO X3 plate reader (PerkinElmer). Nonspecific background values were corrected using the reference absorbance at 690 nm (Nishiga et al, 2017).

## Dual luciferase assays

Full-length PCR fragments of the 3′-UTR of *Fst* were amplified from mouse liver cDNA and subcloned into psiCHECK-2 vectors (Promega). Mutations were introduced into the WT 3′-UTR using a QuikChange II XL Site-Directed Mutagenesis Kit (Agilent) (WT: TGCAAGTCATGT AAAA**ATGCAA**CGCTGTAATGTGGCT, mutant: TGCAAGTCATG TAAA**ATT**C**CT**A**CGCTGTAATGTGGCT; bold letters indicate the miR-33a and miR-33b seed sequences). Luciferase assays were performed as previously described (Horie et al, 2013).

## Adeno-associated virus (AAV) vector construction

AAV vectors were produced using the AAV-2 Helper Free System (Cell Biolabs, Inc. CA) according to the manufacturer's protocol (Horie et al, 2021; Kimura et al, 2019). Plasmids of AAV capsid serotype 9 (pAAV-RC9) were obtained from Penn Vector Core (University of Pennsylvania). AAV-shRNA plasmids containing sequences targeting mouse *Cdk6* (5′-GGATATGATGTTT-CAGCTT-3′), *Abca1* (5′-GAAGAATCTGACATTTCGAAG-3′), and *Fst* (5′-GGAGGATGTGAACGACAAT-3′) mRNA were cloned into pAAV-dsRed-shRNA vectors. For the negative control, pAAV-control-shRNA was constructed using a sequence that was predicted not to target any vertebrate genes, as provided in the BLOCK-iT Pol II miR RNAi Expression Vector Kit (ThermoFisher Scientific). AAV-293 cells (Agilent Technologies, CA, USA) were transfected with pAAV-control-shRNA, pAAV-*Cdk6* shRNA, pAAV-*Abca1* shRNA, or pAAV-*Fst* shRNA; pHelper; or pAAV-RC9 vector plasmids using a PEI MAX-mediated transfection method (Polysciences, Inc.). The medium was replaced 24 h after transfection. The cells were collected using a cell scraper 48 h after medium replacement, resuspended, and frozen −80 °C. For AAV9 vector extraction, the suspensions were frozen at −80 °C for 7 min, thawed in a water bath at 37 °C for 2 min, and vortexed for 1 min. This cycle was repeated four times. Benzonase endonuclease was added, vortexed, and incubated at 45 °C for 15 min. The mixture was centrifuged twice ($18,000 \times g$, 10 min, 4 °C) to remove cell debris, and the supernatant virus solution was stored. To determine the virus titer, a portion of the virus solution was treated with Benzonase and proteinase K followed by viral DNA extraction using phenol/chloroform/isoamyl alcohol. The viral titer was

measured using qPCR targeting the inverted terminal repeat region (forward primer, 5′-GGAACCCCTAGTGATGGAGTT-3′; reverse primer, 5′-CGGCCTCAGTGAGCGA-3′). Serial dilutions of plasmid standard pAAV-GFP ($10^8$, $10^7$, $10^6$, $10^5$, and $10^4$ plasmid copies per 5 µl) were prepared and analyzed in duplicate by qPCR for use as positive controls and in standard curve calculations.

## AAV9 vector administration

miR-33a-KO/*mdx* mice (8 weeks old) were injected in both legs with 50 µl of AAV9-*Cdk6* shRNA, *Abca1* shRNA, *Fst* shRNA, or AAV9-control shRNA viral vectors ($1 \times 10^{12}$ viral particles). They were sacrificed 28 days after AAV vector injection for histological analysis.

## Wire-hanging test

A metal wire (thickness, 2 mm; length, 40 cm) was attached to two vertical posts positioned 30 cm above standard bedding. The longest suspension time method was used. Two trials were performed, and the average hanging time was recorded. The holding impulse was calculated as g × s = body mass (g) × hanging time (s) (Aartsma-Rus and van Putten, 2014).

## Treadmill endurance test

Three days before the treadmill test, the mice underwent preliminary training at 10 m/min for 20 min for habituation. On the test day, after a 5-min warm-up at 5 m/min and a 30-min rest, the test was started at 10 m/min, and the speed was increased by 1 m/min every 2 min. The treadmill was not inclined during any of the phases (Aartsma-Rus and van Putten, 2014; Podkalicka et al, 2020; Yamasaki et al, 2022).

## AMO injection

As described in our previous paper, we generated 12-mer anti-miR-33a and anti-miR-33b oligonucleotides that are complementary to miR-33a and miR-33b, respectively (Yamasaki et al, 2022). To increase the binding affinity of these oligonucleotides to miR-33a and miR-33b, amidated nucleic acids (AmNAs) were used in the synthesis of AMOs with phosphorothioate (PS)-modified linkages. AmNAs also have the effect of reducing liver toxicity. As a control, we generated control AmNAs containing random sequences. AMOs were administered to 8-week-old mice either locally (50 µg injected into the TA muscle weekly for 4 weeks) (Ge et al, 2011) or systemically (subcutaneous injection of 10 or 20 mg/kg bw weekly for 4 weeks) (Miyagawa et al, 2023; Yamasaki et al, 2022).

## RNA sequencing analysis

Total RNA extracted from muscles systemically treated with control AmNA or anti-miR-33b (20 mg/kg bw) was sequenced and analyzed by Macrogen, Inc. Differential expression analysis was performed using likelihood ratio tests and quasi-likelihood F-tests in the edgeR package, and *P* values were adjusted using the Benjamini–Hochberg method. Gene ontology functional analysis was conducted using hypergeometric tests. Pathway analysis was performed using DAVID (Database for Annotation, Visualization,

and Integrated Discovery). Statistical significance was evaluated using modified Fisher's exact tests (EASE score), and $P$ values were adjusted using the Benjamini–Hochberg method. The RNA-seq data have been deposited in the BioStudies database under accession number S-BSST2081.

## Myotube differentiation of iPS cells derived from a patient with DMD

A human iPS cell line derived from a patient with DMD (CiRA00111; deletion of exon 44, dystrophin gene) was generated as previously described (Li et al, 2015). Tet-induced MyoD expression system was introduced into this iPS cell line using piggyBac vector system (Uchimura et al, 2017) and differentiated into myotube as previously described (Uchimura et al, 2017; Yoshida et al, 2017). Cells were seeded at a density of $2.0 \times 10^4$ cells/cm² on six-well plates coated with Matrigel (BD) in StemFit + Y-27632 medium. Following a 24 h and 48 h period, the media were replaced in sequence with PECM (Reprocell) + 10 μM Y-27632 and PECM + 0.3 μg/mL doxycycline (Dox; LKT Labs), respectively. Two days later, the pre-differentiated hiPSCs were reseeded onto new Matrigel-coated 6- to 98-well plates in αMEM (Nacalai) supplemented with 2% horse serum (HS; Sigma), 3 μM Y-27632 and 200 mM 2-mercaptoethanol (2-ME; Nacalai). 3 days after reseeding, the cells were transfected with control-AmNA or anti-miR-33b at a concentration of 100 nM using Lipofectamine RNAiMAX according to the manufacturer's instructions. 6 h after transfection, the medium was replaced with a 2% HS/αMEM solution containing 200 mM 2-ME, 1.0 ug/mL Dox, 5 μM SB431542 (Wako), and 10 ng/ml IGF-1 (PeproTech). After 72 h of transfection, Dox was removed from the medium, and samples were collected 2–4 h later.

### Graphics

Synopsis images were provided by Servier Medical Art (https://smart.servier.com/smart_image/smart-muscle/), NIAID NIH BioArtSource (https://bioart.niaid.nih.gov/bioart/000598), and TogoTV(© 2016 DBCLS TogoTV: https://togotv.dbcls.jp/togopic.2014.60.html; https://togotv.dbcls.jp/togopic.2024.008.html), all licensed under CC BY 4.0 (https://creativecommons.org/licenses/by/4.0/). Images in Fig. 5A were provided by NIAID NIH BioArtSource (https://bioart.niaid.nih.gov/bioart/000372, https://bioart.niaid.nih.gov/bioart/000505), also licensed under CC BY 4.0.

### Statistical analysis

The experiment was conducted and analyzed without prior sample size estimation, and blinding and randomization procedures were not applied. Data were presented as the mean ± standard error of the mean (SEM). Comparisons between two groups were performed using two-sided paired or unpaired $t$-tests. Comparisons between multiple groups were performed using one- or two-way analysis of variance (ANOVA) with Tukey post-hoc test. $P$ values < 0.05 were deemed statistically significant. Analyses were performed using ImageJ 1.44p, GraphPad Prism 6.07 and 8.4.3 (GraphPad Software, Inc.) and Microsoft Excel 2019 (Microsoft Corporation).

**The paper explained**

**Problem**

Muscular dystrophy is a class of genetic diseases characterized by progressive skeletal muscle weakness and degeneration. Among them, Duchenne muscular dystrophy (DMD) is one of the most severe and prevalent forms, caused by mutations in the dystrophin gene. Although recent therapeutic advances such as exon skipping have led to partial restoration of dystrophin in specific subsets of DMD patients and supportive care can mildly slow disease progression, no curative treatments are currently available, and overall therapeutic efficacy remains limited.

**Results**

miR-33a/b were found to play substantial roles in skeletal muscle regeneration. miR-33a-knockout (KO) mice showed accelerated muscle regeneration in response to CTX injury and ameliorated the dystrophic phenotype in *mdx* mice (widely used genetic DMD models). In contrast, as a gain-of-function, miR-33b knock-in (KI) mice exacerbated the dystrophic phenotype in *mdx* mice. The number of Pax7⁺ satellite cells increased in miR-33a-KO mice and decreased in miR-33b-KI mice by targeting *Cdk6*, *Fst*, and partially, *Abca1*. Therapeutic inhibition of miR-33a/b using ASO ameliorated the phenotype of *mdx* mice. Furthermore, miR-33b inhibition upregulated miR-33 target genes in myotubes from iPS cells derived from a patient with DMD.

**Impact**

miR-33a/b act as negative regulators of muscle regeneration by suppressing satellite cell expansion. Therapeutical inhibition of miR-33a/b may be a novel therapeutic approach for patients with muscular dystrophy, including DMD.

## Data availability

The data supporting our study findings are available from the corresponding authors upon reasonable request. Source data are included in this published article. The RNA-seq data have been deposited in the BioStudies database under accession number S-BSST2081.

The source data of this paper are collected in the following database record: biostudies:S-SCDT-10_1038-S44321-025-00273-9.

## Peer review information

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

## Acknowledgements

The patient-derived iPS cell line used in this study was kindly provided by Dr. Hidetoshi Sakurai from the Center for iPS Cell Research and Application (CiRA), Kyoto University. This work was supported by the Ministry of Education, Culture, Sports, Science, and Technology (MEXT), the Japan Society for the Promotion of Science (JSPS) KAKENHI grants 17K09860, 20K08904, 23K07988 (TH) and 17H04177, 20H03675, and 23K27595 (KOn), and the Japan Science and Technology Agency for the Fusion Oriented Research for Disruptive Science and Technology program (JST FOREST) grant JPMJFR235F (TH). This work was also supported by AMED under grants 21ym0126013h0001, 23ym0126074h0002 (KOn), 24ek0210202h0001 (TH), and 24am0521009 (SO), as well as grants from Daiwa Securities Foundation, Takeda Science Foundation, and Ono Medical Science Foundation (TH).

## Author contributions

**Naoya Sowa**: Data curation; Formal analysis; Investigation. **Takahiro Horie**: Conceptualization; Data curation; Formal analysis; Funding acquisition; Investigation; Writing—original draft; Writing—review and editing. **Yuya Ide**: Investigation. **Osamu Baba**: Data curation; Investigation. **Kengo Kora**: Investigation. **Takeshi Yoshida**: Resources; Investigation. **Yujiro Nakamura**: Investigation. **Shigenobu Matsumura**: Investigation. **Kazuki Matsushita**: Investigation. **Miyako Imanaka**: Investigation. **Fuquan Zou**: Investigation. **Eitaro Kume**: Investigation. **Hidenori Kojima**: Investigation. **Qiuxian Qian**: Investigation. **Kayo Kimura**: Investigation. **Ryotaro Otsuka**: Investigation. **Noriko Hara**: Investigation. **Tomohiro Yamasaki**: Investigation. **Chiharu Otani**: Investigation. **Yuta Tsujisaka**: Investigation. **Tomohide Takaya**: Resources; Investigation. **Chika Nishimura**: Resources; Investigation. **Dai Watanabe**: Resources; Investigation. **Koji Hasegawa**: Data curation; Investigation. **Jun Kotera**: Data curation; Investigation. **Kozo Oka**: Data curation; Investigation. **Ryo Fujita**: Data curation; Investigation. **Akihiro Takemiya**: Resources. **Takashi Sasaki**: Data curation; Investigation. **Yuuya Kasahara**: Resources. **Satoshi Obika**: Resources; Data curation; Funding acquisition. **Takeshi Kimura**: Supervision. **Koh Ono**: Conceptualization; Data curation; Formal analysis; Supervision; Funding acquisition; Writing—original draft; Project administration; Writing—review and editing.

Source data underlying figure panels in this paper may have individual authorship assigned. Where available, figure panel/source data authorship is listed in the following database record: biostudies:S-SCDT-10_1038-S44321-025-00273-9.

## Disclosure and competing interests statement

JK, KOk, RF, AT, and TS are employees of Mitsubishi Tanabe Pharma Corporation.

