## [Peer Review File · EMBO Molecular Medicine]

MicroRNA-33 inhibition ameliorates muscular dystrophy by enhancing skeletal muscle regeneration

Naoya Sowa, Takahiro Horie, Yuya Ide, Osamu Baba, Kengo Kora, Takeshi Yoshida, Yujiro Nakamura, Shigenobu Matsumura, Kazuki Matsushita, Miyako Imanaka, Fuquan Zou, Eitaro Kume, Hidenori Kojima, Qiuxian Qian, Kayo Kimura, Ryotaro Otsuka, Noriko Hara, Tomohiro Yamasaki, Chiharu Otani, Yuta Tsujisaka, Tomohide Takaya, Chika Nishimura, Dai Watanabe, Koji Hasegawa, Jun Kotera, Kozo Oka, Ryo Fujita, Akihiro Takemiya, Takashi Sasaki, Yuuya Kasahara, Satoshi Obika, Takeshi Kimura, and Koh Ono

Corresponding authors: Koh Ono (kohono@kuhp.kyoto-u.ac.jp) , Takahiro Horie (thorie@kuhp.kyoto-u.ac.jp)

Review Timeline:

Submission Date:	27th Feb 25
Editorial Decision:	14th Mar 25
Revision Received:	12th Jun 25
Editorial Decision:	20th Jun 25
Revision Received:	30th Jun 25
Accepted:	4th Jul 25

Editor: Zeljko Durdevic

Transaction Report:

14th Mar 2025

Dear Dr. Ono,

Thank you for the submission of your manuscript to EMBO Molecular Medicine. We have now received feedback from the two reviewers who agreed to evaluate your manuscript. Both referees recognize interest of the study but also raise important concerns that should be addressed in a major revision. If you would like to discuss further the points raised by the referees, I am available to do so via email or video. Let me know if you are interested in this option.

We would welcome the submission of a revised version within three months for further consideration. Please let us know if you require longer to complete the revision.

I look forward to receiving your revised manuscript.

Yours sincerely,

Zeljko Durdevic

Zeljko Durdevic
Senior Editor
EMBO Molecular Medicine

We require:

- 1) A .docx formatted version of the manuscript text (including legends for main figures, EV figures and tables). Please make sure that the changes are highlighted to be clearly visible.
- 2) Individual production quality figure files as .eps, .tif, .jpg (one file per figure). For guidance, download the 'Figure Guide PDF': (<https://www.embopress.org/page/journal/17574684/authorguide#figureformat>).
- 3) A .docx formatted letter INCLUDING the reviewers' reports and your detailed point-by-point responses to their comments. As part of the EMBO Press transparent editorial process, the point-by-point response is part of the Review Process File (RPF), which will be published alongside your paper.
- 4) A complete author checklist, which you can download from our author guidelines (<https://www.embopress.org/page/journal/17574684/authorguide#submissionofrevisions>). Please insert information in the checklist that is also reflected in the manuscript. The completed author checklist will also be part of the RPF.
- 5) Please note that all corresponding authors are required to supply an ORCID ID for their name upon submission of a revised manuscript.
- 6) It is mandatory to include a 'Data Availability' section after the Materials and Methods. Before submitting your revision, primary datasets produced in this study need to be deposited in an appropriate public database, and the accession numbers and

database listed under 'Data Availability'. Please remember to provide a reviewer password if the datasets are not yet public (see <https://www.embopress.org/page/journal/17574684/authorguide#dataavailability>).

12) Author contributions: You will be asked to provide CRediT (Contributor Role Taxonomy) terms in the submission system. These replace a narrative author contribution section in the manuscript.

13) A Conflict of Interest statement should be provided in the main text.

14) Every published paper now includes a 'Synopsis' to further enhance discoverability. Synopses are displayed on the journal webpage and are freely accessible to all readers. They include a short stand first (maximum of 300 characters, including space) as well as 2-5 one-sentences bullet points that summarizes the paper. Please write the bullet points to summarize the key NEW findings. They should be designed to be complementary to the abstract - i.e. not repeat the same text. We encourage inclusion of key acronyms and quantitative information (maximum of 30 words / bullet point). Please use the passive voice. Please attach

these in a separate file or send them by email, we will incorporate them accordingly.

15) Include a Reagents and Tools Table as part of the Methods section, which can be downloaded from our author guidelines (<https://www.embopress.org/page/journal/17574684/authorguide#structuredmethods>)

***** Reviewer's comments *****

Referee #1 (Comments on Novelty/Model System for Author):

The models used are appropriate for evaluating the ability of manipulation of miR-33a/b expression in mdx mice. The suitability of this study for the journal is high as it will be useful for the DMD therapeutics and biomarker fields.

Referee #1 (Remarks for Author):

The manuscript by Sowa et al., centers on the evaluation of a microRNA, miR-33a/b, and its role in skeletal muscle regeneration. MicroRNA-33a was shown to be induced in expression in myogenic differentiation and in the mdx muscles, a model of Duchenne muscular dystrophy (DMD). miR-33a deficiency enhanced muscle regeneration response to cardiotoxin injury and attenuated muscle degeneration and fibrosis in mdx mice. Humanized mice overexpressing miR-33a/b showed exacerbated muscle degeneration and fibrosis. miR-33a/b inhibited satellite cell proliferation, leading to reduced muscle regeneration and increased fibrosis by targeting Cdk6, Fst, and Abca1 RNA transcripts. Administration of anti-miRNA oligonucleotides targeting miR-33a/b ameliorated the dystrophic phenotype in mdx mice. The authors concluded that miR-33a/b suppression or inhibition may therapeutically benefit DMD muscle pathologies.

This is an overall well-written manuscript, with the data and experiments thoroughly analyzed and experimentally justified. The experiments are appropriately powered and statistical analysis is evident. Most of my comments are minor and are focused on clarification of key experimental details. This is a comprehensive manuscript in scope and finding. It would likely be an important addition to the microRNA therapeutics and DMD biology fields.

Overall Comments:

1. More details should be included on the nature of the AMOs for miR-33a/b inhibition. What was the chemical backbone of the AMOs? What is the turnover or longevity of the AMOs in mice?
2. Has it been demonstrated how and by what mechanism does miR-33a/b regulate cell cycle progression? Have the authors compared their results in expression analysis to entire cell cycle targets identified (CCND1) from the Cirera-Salinas 2012 manuscript in their transcriptomic analysis?
3. Did the authors evaluate the hearts or cardiac function in the miR-33a/b KO/mdx mice, the AMO-miR-33a-treated mdx, or the miR-33 KI mice? This seems like a missed opportunity to determine if the effects of miR-33a/b inhibition were systemic.
4. It might be helpful for the authors to compare their anti-miR-33a qPCR analysis with that of an mdx or DMD patient RNA-seq transcriptome that is publicly available to see if target transcripts (Abca1, Fst, and Cdk6) change in expression. Minor comment and suggestion.

Referee #2 (Remarks for Author):

The manuscript by Sowa et al. investigates the role of miR-33 in the muscle regeneration process of mdx mice, a genetic model of Duchenne muscular dystrophy (DMD). The study demonstrates that miR-33 inhibits muscle regeneration in mdx mice and explores the therapeutic potential of anti-miR-33 oligonucleotides in ameliorating dystrophic symptoms. This research holds significant clinical promise and may provide new therapeutic directions for patients with DMD. However, several issues related to experimental design, methodologies, and data interpretation need to be addressed:

1. Supplementary Figure 2E: The statistical analysis does not support the statement in the main text: "Muscle weight tended to increase after CTX injection, showing a significant difference between miR-33a-KO mice and WT mice at 21 days after CTX injection (Supplementary Fig. 2E)." The authors could have mislabeled figure. Please clarify this discrepancy and provide appropriate statistical evidence to support the claim.
2. Supplementary Figures 3A and 3B: While the protein levels of CDK6 show differences between groups, the RNA levels do not display similar variation. The figure legend states that "Cdk6 is a target of miR-33." The authors may need to make a comment that miR-33 seems to regulate CDK6 at the translation rather than affecting mRNA stability.
3. Immunofluorescence Imaging: The immunofluorescence images presented in Figures 2A and 4C exhibit high background fluorescence, making it challenging to distinguish positive cells from non-positive ones. Please clarify the criteria used to identify positive cells in these images. If specific thresholds or selection methods were applied, please include detailed descriptions in the Methods section.
4. Quantification of Positive Cells: There appears to be inconsistency between method and figure legend in the quantification of positive cells from immunofluorescence images. In Figure 2B, the Methods section states: "Three random images per skeletal muscle were obtained from each mouse to quantify the Ki-67-, Pax7-, and MyoD-positive cells." However, while the figure legend indicates n = 5 per group, the graphs display 19 data points per group, which is confusing. Please revisit the experiments and provide a clear explanation of how the data were collected and analyzed.
5. Extent of miR-33 Contribution to DMD Pathology: It remains unclear to what extent miR-33 contributes to the overall pathology of DMD and the degree to which motor function can be restored through anti-miR-33 treatment. Including wild-type (WT) mice as an additional comparison group would provide valuable context on this issue.
6. Supplementary Figure 11G: In the human data, a statistical difference is also observed between the miR-control and miR-33 in the mutant group. The authors need to give a brief comment on it.

Response to Reviewer #1

We are grateful to Reviewer #1 for the informative and useful comments. As described below, we have considered all of these comments and used them to improve our manuscript.

Referee #1 (Remarks for Author):

The manuscript by Sowa et al., centers on the evaluation of a microRNA, miR-33a/b, and its role in skeletal muscle regeneration. MicroRNA-33a was shown to be induced in expression in myogenic differentiation and in the mdx muscles, a model of Duchenne muscular dystrophy (DMD). miR-33a deficiency enhanced muscle regeneration response to cardiotoxin injury and attenuated muscle degeneration and fibrosis in mdx mice. Humanized mice overexpressing miR-33a/b showed exacerbated muscle degeneration and fibrosis. miR-33a/b inhibited satellite cell proliferation, leading to reduced muscle regeneration and increased fibrosis by targeting Cdk6, Fst, and Abca1 RNA transcripts. Administration of anti-miRNA oligonucleotides targeting miR-33a/b ameliorated the dystrophic phenotype in mdx mice. The authors concluded that miR-33a/b suppression or inhibition may therapeutically benefit DMD muscle pathologies.

This is an overall well-written manuscript, with the data and experiments thoroughly analyzed and experimentally justified. The experiments are appropriately powered and statistical analysis is evident. Most of my comments are minor and are focused on clarification of key experimental details. This is a comprehensive manuscript in scope and finding. It would likely be an important addition to the microRNA therapeutics and DMD biology fields.

Overall Comments:

- 1. More details should be included on the nature of the AMOs for miR-33a/b inhibition.**

What was the chemical backbone of the AMOs? What is the turnover or longevity of the AMOs in mice?

We thank the reviewer for highlighting this important point. As described in our previous paper, we generated 12-mer anti-miR-33a and anti-miR-33b oligonucleotides that are complementary to miR-33a and miR-33b, respectively. To increase the binding affinity of these oligonucleotides to miR-33a and miR-33b, amidated nucleic acids (AmNAs) were used in the synthesis of AMOs with phosphorothioate (PS)-modified linkages. AmNAs also has the effect of reducing liver toxicity. As a control, we generated AmNAs containing random sequences. The inhibitory activities of the AMOs

were evaluated using a dual luciferase assay.

Regarding the turnover rate or longevity of AMOs in mice, this study administered the drug once a week, but other MASH models (Life Sci Alliance. 2023 Jun 1;6(8):e202301902.) have demonstrated efficacy with administration once every 2 weeks.

We have added information about AMO modifications to the main text as follows.

Inserted sentences (on page 26, paragraph 1, lines 747-752): As described in our previous paper, we generated 12-mer anti-miR-33a and anti-miR-33b oligonucleotides that are complementary to miR-33a and miR-33b, respectively (Sci Rep 2022). To increase the binding affinity of these oligonucleotides to miR-33a and miR-33b, amidated nucleic acids (AmNAs) were used in the synthesis of AMOs with phosphorothioate (PS)-modified linkages. AmNAs also have the effect of reducing liver toxicity. As a control, we generated control AmNAs containing random sequences.

2. Has it been demonstrated how and by what mechanism does miR-33a/b regulate cell cycle progression? Have the authors compared their results in expression analysis to entire cell cycle targets identified (CCND1) from the Cirera-Salinas 2012 manuscript in their transcriptomic analysis?

Thank you very much for the comment. As shown in Supplemental Figure 3A, there was no change in the protein level of CCND1, and we believe that the change in CDK6 rather influenced the results. Furthermore, this result is supported by CDK6 knockdown experiments (Figure 5 and Supplementary Figure 7).

3. Did the authors evaluate the hearts or cardiac function in the miR-33a/b KO/mdx mice, the AMO-miR-33a-treated mdx, or the miR-33 KI mice? This seems like a missed opportunity to determine if the effects of miR-33a/b inhibition were systemic.

We appreciate the reviewer for the important questions about the cardiac functions of mice. As previously

reported, cardiac fibrosis is not advanced in young Mdx mice (2001 Jun;50(3):509-15). In our

experiments, myocardial fibrosis was generally mild, and there was a tendency for it to decrease in the absence of miR-33a/b; however, no significant difference was observed between WT/mdx and miR-33aKO/mdx (Figure A). Similarly, there was a tendency for an increase between WT/mdx and miR-33b-KI/mdx, but no significant difference was observed (Figure B).

4. It might be helpful for the authors to compare their anti-miR-33a qPCR analysis with that of an mdx or DMD patient RNA-seq transcriptome that is publicly available to see if target transcripts (Abca1, Fst, and Cdk6) change in expression. Minor comment and suggestion.

Thank you very much for the valuable comments. Urinary stem cells (USCs) have garnered attention as a non-invasive, convenient, and low-cost cell source for human disease research (HGG Adv. 2021 Aug 24;3(1):100054). In this study, we demonstrate changes in the expression of Abca1, Fst, and Cdk6 in USCs obtained from i) three healthy donors (C-n) and ii) patients with DMD (IG, exon 45 deletion associated) (IG-n). In patients with DMD, the gene expression of Abca1 and Cdk6 is reduced, suggesting that these genes may function as therapeutic targets. On the other hand, no significant

changes in Fst were observed between healthy donors and patients with DMD; however, if its expression increases upon suppression of miR-33a/b, it may contribute to disease improvement through its myokine function (Figure C).

Response to Reviewer #2

We are grateful to Reviewer #2 for the informative and useful comments. As described below, we have considered all of these comments and used them to improve our manuscript.

Referee #2 (Remarks for Author):

The manuscript by Sowa et al. investigates the role of miR-33 in the muscle regeneration process of mdx mice, a genetic model of Duchenne muscular dystrophy (DMD). The study demonstrates that miR-33 inhibits muscle regeneration in mdx mice and explores the therapeutic potential of anti-miR-33 oligonucleotides in ameliorating dystrophic symptoms. This research holds significant clinical promise and may provide new therapeutic directions for patients with DMD. However, several issues related to experimental design, methodologies, and data interpretation need to be addressed:

1. Supplementary Figure 2E: The statistical analysis does not support the statement in the main text: "Muscle weight tended to increase after CTX injection, showing a significant difference between miR-33a-KO mice and WT mice at 21 days after CTX injection (Supplementary Fig. 2E)." The authors could have mislabeled figure. Please clarify this discrepancy and provide appropriate statistical evidence to support the claim.

We appreciate the reviewer for pointing out the confusion. The columns in the figure were difficult to understand and did not allow for appropriate comparisons. Regarding Supplementary Figure 2E, we have changed the left figure to the right figure (Figure D).

Figure D

We have revised the sentence as follows.

Revised sentence (on page 7, paragraph 1, lines 172-173): Muscle weight tended to increase after CTX injection, showing a significant difference in miR-33a-KO mice at 21 days after CTX injection (Supplementary Fig. 2E).

2. Supplementary Figures 3A and 3B: While the protein levels of CDK6 show differences between groups, the RNA levels do not display similar variation. The figure legend states that

"Cdk6 is a target of miR-33." The authors may need to make a comment that miR-33 seems to regulate CDK6 at the translation rather than affecting mRNA stability.

Thank you very much for the valuable comment. As pointed out, the protein levels of CKD6 changed, whereas the mRNA levels were not changed. We have added the following sentence to the main text.

Inserted sentences (on page 7, paragraph 1, lines 183-184): Therefore, miR-33 appears to regulate CDK6 at the translational level rather than affecting mRNA stability.

3. Immunofluorescence Imaging: The immunofluorescence images presented in Figures 2A and 4C exhibit high background fluorescence, making it challenging to distinguish positive cells from non-positive ones. Please clarify the criteria used to identify positive cells in these images. If specific thresholds or selection methods were applied, please include detailed descriptions in the Methods section.

Thank you very much for your comments. In the immunostaining, we counted the positive cells from three random images as described in the Materials and methods section. For the determination of positive cells, we referred to previous papers (Hindi & Kumar, 2016; Podkalicka et al., 2020; Wu et al., 2015). Identical brightness and contrast settings were applied to all images to maintain consistency.

We have added the following sentences in the methods section.

Inserted sentences (on page 23, paragraph 2, lines 674-677): The images were captured using an Axio Observer microscope (Zeiss) and analyzed using ImageJ 1.44p software. To ensure consistency across the dataset, identical brightness and contrast parameters were applied to all images.

4. Quantification of Positive Cells: There appears to be inconsistency between method and figure legend in the quantification of positive cells from immunofluorescence images. In Figure 2B, the Methods section states: "Three random images per skeletal muscle were obtained from each mouse to quantify the Ki-67-, Pax7-, and MyoD-positive cells." However, while the figure legend indicates n = 5 per group, the graphs display 19 data points per group, which is confusing. Please revisit the experiments and provide a clear explanation of how the data were collected and analyzed.

Thank you very much for pointing out the inconsistency of the quantification of positive cells between method and figure legends. We have noted the Ki-67-, Pax7-, and MyoD-positive cells, but

there was an error in the MyoD-, Pax7-, and DAPI-positive cells. Additionally, regarding the number of data points in the figure, we have

standardized the data

by randomly selecting three images after staining and corrected it to 15 data points obtained from five individuals. Therefore, we will revise Figure 2B by moving it from the left to the right in Figure E.

We have revised the sentences as follows.

Revised sentences (on page 23, paragraph 2, lines 671-674): Three random images per skeletal muscle were obtained from each mouse to quantify Ki-67-, Pax7-, and DAPI-positive cells, and Pax7-negative or positive, MyoD-negative or positive, and DAPI-positive cells (i.e., Ki-67⁺/Pax7⁺, Pax7⁻/MyoD⁺, Pax7⁺/MyoD⁺, and Pax7⁺/MyoD⁻).

5. Extent of miR-33 Contribution to DMD Pathology: It remains unclear to what extent miR-33 contributes to the overall pathology of DMD and the degree to which motor function can be restored through anti-miR-33 treatment. Including wild-type (WT) mice as an additional comparison group would provide valuable context on this issue.

We appreciate the valuable comments and suggestions to use WT mice as a comparison.

In WT mice, fibrosis does not occur in skeletal muscle; therefore, the extent to which miR-33 inhibition demonstrates histological improvement can be determined from the results of this study.

On the other hand, regarding motor function, we simultaneously ran WT mice and Mdx mice on a treadmill

at 8 weeks of age. The results showed that WT mice ran approximately twice the distance of Mdx

mice (Figure F). Comparing this with miR-33a KO data, it is inferred that in the absence of miR-33a from birth, the running ability of Mdx mice at 8 weeks is closer to that of WT mice. On the other hand, injecting anti-miR-33b for 4 weeks starting at 8 weeks of age suppresses the decline in running performance in miR-33b KI/mdx mice, but does not improve running performance to the level of WT mice at 8 weeks of age (Fig. 8B). Therefore, it is expected that the difference from WT mice will further increase even in anti-miR-33b-injected group.

6. Supplementary Figure 11G: In the human data, a statistical difference is also observed between the miR-control and miR-33 in the mutant group. The authors need to give a brief comment on it.

Thank you very much for the comment.

For this 3'UTR experiment, we repeatedly conducted experiments using the same concentration of miR-33 as in the original experiment (Figure G, left) and experiments using twice the concentration (Figure F, right). Since both experiments yielded the same results, we adopted the results of the experiment on the left as Supplementary Figure 11G.

Figure G

20th Jun 2025

Dear Dr. Ono,

Thank you for the submission of your revised manuscript to EMBO Molecular Medicine. I am pleased to inform you that we will be able to accept your manuscript pending the following final amendments:

1) Authors: We note name discrepancies in our submission system and in the manuscript. Kengo Kora in the manuscript and Kengo Koura in our system; Yuta Tsujisaka in the manuscript and Yuta Tujisaka in our system. Please correct.

2) Please address all comments suggested by our data editors listed below:

o Figure legends:

1. Please note that the exact p values are not provided in the legends of figures 2G, 5F, 7A, 8F, 9G.

2. Please indicate the statistical test used for data analysis in the legends of figures 9B, C.

- Rename "Materials and Methods" to "Methods".

- Remove "Material list" from the Methods.

- In Methods, add the following paragraph:

Graphics:

(some of the... OR Figure #... OR synopsis) Graphics were created with BioRender.com.

- Rename "Competing interest" to "Disclosure and competing interests statement". We updated our journal's competing interests policy in January 2022 and request authors to consider both actual and perceived competing interests. Please review the policy <https://www.embopress.org/competing-interests> and update your competing interests if necessary.

- Author contributions: Please remove it from the manuscript and specify author contributions in our submission system. CRediT has replaced the traditional author contributions section because it offers a systematic machine-readable author contributions format that allows for more effective research assessment. Please use the free text boxes beneath each contributing author's name to add specific details on the author's contribution. More information is available in our guide to authors:

<https://www.embopress.org/page/journal/17574684/authorguide#authorshipguidelines>

- Indicate in legends exact n and exact p values, not a range, along with the statistical test used. To keep the figures "clear" some authors found providing an Appendix table Sx with all exact p-values preferable. You are welcome to do this if you want to.

- Place Table 1 after the figure legends.

3) Appendix:

- Add table of content with page numbers in the title page.

- We note 25 cases of blot and image reuse in appendix figures. For example, Appendix Figure S4 panels D, E and F are reused in Appendix Figure S5 panels B, C and D; Appendix Figure S6, panels A, B (images) C, D and E (magnifications) are reused in Appendix Figure S7 panels A, B, C, D and G; Appendix Figure S6 panels H and I are reused in Appendix Figure S8 panels L and M. Please revise all Appendix Figures and omit presentation of the same results in multiple figures. Also, pay attention that the figure legends correspond to the results presented in the figure. Please make sure that the callouts in the main manuscript text are correct after the revision of Appendix Figures.

- Please remove Appendix Table 4 and the reference to it form Reagents Table as bot contain the sane information. Also, rename Appendix Tables to Appendix Table S1 etc. and update their callouts in the main manuscript text

4) The Paper Explained: Please add it to the main manuscript text. Please change subheadings to "Problem", "Results" and "Impact".

5) Synopsis:

- Synopsis text: Please upload it as a separate .doc file.

- Synopsis image: Please resize the image to 550 px-wide x 300-600 pixels high and upload it as a high-resolution jpeg file.

6) Source data: Please upload source data as one zipped folder per figure.

7) As part of the EMBO Publications transparent editorial process initiative (see our Editorial at <http://embomolmed.embopress.org/content/2/9/329>), EMBO Molecular Medicine will publish online a Review Process File (RPF) to accompany accepted manuscripts. This file will be published in conjunction with your paper and will include the anonymous referee reports, your point-by-point response and all pertinent correspondence relating to the manuscript. Let us know whether you agree with the publication of the RPF and as here, if you want to remove or not any figures from it prior to publication. Please note that the Authors checklist will be published at the end of the RPF.

8) Please provide a point-by-point letter INCLUDING my comments as well as the reviewer's reports and your detailed responses (as Word file).

I look forward to reading a new revised version of your manuscript as soon as possible.

Yours sincerely,

Zeljko Durdevic

Zeljko Durdevic
Senior Editor
EMBO Molecular Medicine

*** Instructions to submit your revised manuscript ***

- 1) a .docx formatted version of the manuscript text (including Figure legends and tables)
- 2) Separate figure files*
- 3) supplemental information as Expanded View and/or Appendix. Please carefully check the authors guidelines for formatting Expanded view and Appendix figures and tables at <https://www.embopress.org/page/journal/17574684/authorguide#expandedview>
- 4) a letter INCLUDING the reviewer's reports and your detailed responses to their comments (as Word file).
- 5) The paper explained: EMBO Molecular Medicine articles are accompanied by a summary of the articles to emphasize the major findings in the paper and their medical implications for the non-specialist reader. Please provide a draft summary of your article highlighting
 - the medical issue you are addressing,
 - the results obtained and
 - their clinical impact.This may be edited to ensure that readers understand the significance and context of the research. Please refer to any of our published articles for an example.
- 6) Author contributions: the contribution of every author must be detailed in a separate section.
- 7) EMBO Molecular Medicine now requires a complete author checklist (<https://www.embopress.org/page/journal/17574684/authorguide>) to be submitted with all revised manuscripts. Please use the checklist as guideline for the sort of information we need WITHIN the manuscript. The checklist should only be filled with page numbers where the information can be found. This is particularly important for animal reporting, antibody dilutions (missing) and exact values and n that should be indicated instead of a range.
- 8) Every published paper now includes a 'Synopsis' to further enhance discoverability. Synopses are displayed on the journal webpage and are freely accessible to all readers. They include a short stand first (maximum of 300 characters, including space) as well as 2-5 one sentence bullet points that summarise the paper. Please write the bullet points to summarise the key NEW findings. They should be designed to be complementary to the abstract - i.e. not repeat the same text. We encourage inclusion of key acronyms and quantitative information (maximum of 30 words / bullet point). Please use the passive voice. Please attach these in a separate file or send them by email, we will incorporate them accordingly.

You are also welcome to suggest a striking image or visual abstract to illustrate your article. If you do please provide a jpeg file 550 px-wide x 300-600px high.

9) A Conflict of Interest statement should be provided in the main text

10) Please note that we now mandate that all corresponding authors list an ORCID digital identifier. This takes <90 seconds to complete. We encourage all authors to supply an ORCID identifier, which will be linked to their name for unambiguous name identification.

Currently, our records indicate that the ORCID for your account is 0000-0002-4163-980X.

Link Not Available

11) Include a Reagents and Tools Table as part of the Methods section, which can be downloaded from our author guidelines (<https://www.embopress.org/page/journal/17574684/authorguide#structuredmethods>)

Photos 400-800 DPI

*Additional important information regarding figures and illustrations can be found at

<https://bit.ly/EMBOPressFigurePreparationGuideline>. See also figure legend preparation guidelines:

<https://www.embopress.org/page/journal/17574684/authorguide#figureformat>

***** Reviewer's comments *****

Referee #1 (Comments on Novelty/Model System for Author):

The models generated a novel and evaluate the consequences of miR-33a ablation on the DMD/mdx background.

Referee #1 (Remarks for Author):

The authors provided a comprehensive response to my previous comments and the other reviewers'. Comprehensive analysis of the miR-33 targets are also provided as well as alternative discussion of the data interpretation(s). No concerns further noted.

Referee #2 (Comments on Novelty/Model System for Author):

The revised manuscript has been significantly improved. I think it is suitable for publication.

Referee #2 (Remarks for Author):

The authors have satisfactorily addressed my comments. I have no further comments.

The authors addressed the remaining editorial issues.

4th Jul 2025

Dear Dr. Ono,

We are pleased to inform you that your manuscript is accepted for publication and is now being sent to our publisher to be included in the next available issue of EMBO Molecular Medicine. Please note that data deposited in public repositories must be freely accessible at the time of publication.

Zeljko Durdevic
Senior Editor
EMBO Molecular Medicine
